# HypoSpace: A Diagnostic Benchmark for Set-Valued Hypothesis Generation under Underdetermination and Sublinear Coverage Bounds

**Tingting Chen** [1]  **Beibei Lin** [1]  **Zifeng Yuan** [1]  **Qiran Zou** [1]  **Hongyu He** [1]
**Anirudh Goyal** [2]  **Yew-Soon Ong** [3][4]  **Dianbo Liu** [1]

## Abstract

Many scientific problems are underdetermined: multiple distinct hypotheses are equally consistent with the same observations. In such settings, effective inference requires not only producing valid explanations, but also systematically exploring and covering the admissible hypothesis set. We introduce **HypoSpace**, a benchmark that treats large language models (LLMs) as samplers over finite hypothesis spaces and evaluates them on three metrics: Validity, Uniqueness, and Recovery. HypoSpace spans three structured domains (causal graph inference, gravity-constrained 3D voxel reconstruction, and Boolean genetic interaction modeling) with deterministic validators and exactly enumerable solution spaces, plus real-world anchored case studies. Empirically, HypoSpace reveals a capability- and scale-dependent coverage failure: models can maintain high Validity while exhibiting reduced Uniqueness and Recovery as admissible hypothesis spaces become larger or more combinatorial. We further show that the analysis on stratified decoding partially mitigates this collapse, demonstrating HypoSpace's utility as a diagnostic benchmark for set-valued inference. Code is available at: https://github.com/CTT-Pavilion/_HypoSpace.

## 1. Introduction

Many scientific inference problems are *underdetermined* (Van Fraassen, 1980; Stanford, 2010): the same

observations admit multiple, mechanistically distinct explanations. EEG source imaging is a canonical example—infinitely many neural source distributions produce identical scalp potentials (Michel & Brunet, 2019). Under such conditions, a capable scientific reasoning system should not stop at finding one valid explanation, but systematically explore the space of alternatives.

LLMs are increasingly used to generate scientific hypotheses, making this exploratory capacity essential. Yet current benchmarks reward single-answer correctness (Shojaee et al., 2025; Koblischke et al., 2025; Shojaee et al., 2024; Wang et al., 2024b; Coignion et al., 2024; Hendrycks et al., 2020), leaving untested whether models can enumerate multiple valid hypotheses. We therefore ask: *Can LLMs systematically explore hypothesis spaces under underdetermination?*

To address this gap, we introduce **HypoSpace**, a diagnostic suite for evaluating hypothesis space exploration. We define three complementary metrics for hypothesis generation: **Validity** enforces appropriateness by measuring consistency with observations; **Uniqueness** quantifies originality through non-redundancy among all proposals; and **Recovery** operationalizes fluency by measuring coverage of the enumerated admissible set $\mathcal{H}_O$. Our framework provides deterministic validators and enumerated ground truth, eliminating rater subjectivity and enabling precise measurement.

Our approach treats LLMs as samplers that generate finite sets of candidate hypotheses (Figure 1a). For each problem instance, we enumerate the complete valid set $\mathcal{H}_O$, apply exact validity checks, and assess non-redundancy using task-specific canonicalizers that collapse semantically equivalent forms. By repeatedly sampling and measuring behavior along Validity (VR), Uniqueness (NR), and Recovery (RR), we decouple *being correct* from *exploring comprehensively*—a distinction obscured by traditional correctness-only metrics.

We instantiate this diagnostic suite across three structured domains that mirror scientific inference while enabling exact enumeration of valid hypothesis spaces: **Causal graphs inference**, where models infer all DAGs consistent with single-

---

[1]National University of Singapore, Singapore [2]Meta Superintelligence Labs [3]College of Computing and Data Science, Nanyang Technological University, Singapore [4]Centre for Frontier AI Research (CFAR), Agency for Science, Technology and Research, Singapore. Correspondence to: Dianbo Liu <dianbo@nus.edu.sg>.

*Proceedings of the 43rd International Conference on Machine Learning*, Seoul, South Korea. PMLR 306, 2026. Copyright 2026 by the author(s).

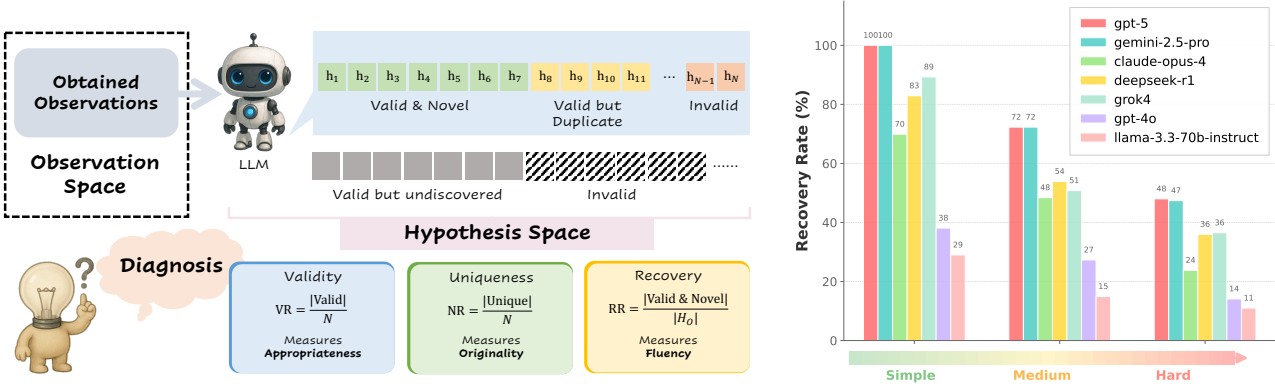

*(a)* HypoSpace Evaluation Framework.

*(b)* Model Comparison on RR.

*Figure 1.* **HypoSpace evaluation framework and model performance comparison.** (a) Our diagnostic approach treats LLMs as samplers over hypothesis spaces. Given observations $O$, models generate $N$ hypotheses that are validated for consistency with data and deduplicated for uniqueness. We measure three complementary indicators: Validity (VR: precision of valid hypotheses), Uniqueness (NR: non-redundancy among proposals), and Recovery (RR: coverage of the enumerated admissible set $\mathcal{H}_O$). (b) Recovery Rate comparison across models on task of Boolean genetic interactions, showing systematic degradation as hypothesis spaces grow from simple to hard settings, with reasoning models generally outperforming non-reasoning models. Higher difficulty corresponds to larger admissible sets.

node intervention observations; **3D voxel reconstruction under gravity**, where models reconstruct spatial configurations from top-down projections while satisfying physical constraints; and **Boolean genetic interactions**, where models propose expressions relating phenotype observations to underlying Boolean programs. Each domain provides natural difficulty scaling through controllable parameters (node counts, grid dimensions, operator complexity) that systematically vary the size of the admissible set $|\mathcal{H}_O|$. This controlled enumeration enables three key capabilities: direct measurement of hypothesis space coverage, precise calibration of task difficulty, and systematic analysis of sampling behavior including mode collapse patterns. We further validate on real-world genetic data (Section 7), where over 100 valid hypotheses remain consistent with observations, demonstrating real-world underdetermination, and LLMs exhibit the same mode collapse seen in synthetic settings. Our goal is diagnostic measurement rather than leaderboard optimization—we seek to understand and improve how models explore solution spaces under underdetermination.

Evaluation across recent instruction-tuned and reasoning-focused LLMs reveals a consistent and concerning pattern: while models often maintain high **Validity** rates when generating admissible hypotheses, they exhibit pronounced *mode collapse* as hypothesis spaces grow. Both **Uniqueness** and **Recovery** degrade predictably with increasing $|\mathcal{H}_O|$ (Figure 1b), indicating that current models tend to circle a small subset of admissible explanations rather than systematically explore the complete space that observations allow. Crucially, because our valid sets are exactly enumerated, these coverage failures are measurable rather than anecdotal, and persist even when traditional accuracy metrics suggest

strong performance. Furthermore, in Section 4, we provide a simple theoretical explanation for why mode collapse occurs on peaked generators. The primary contributions of this work are:

1. **Theoretical formulation:** We frame the evaluation of LLMs' ability to infer multiple distinct hypotheses fitting the same observations as *set-valued inference* under underdetermination, introducing three *diagnostic indicators* that systematically separate correctness from exploration capabilities. To the best of our knowledge, this is the first systematic framework in this direction.

2. **Controlled diagnostic suite:** Three structured tasks with exact *enumeration* of valid hypothesis spaces, enabling non-LLM validity checking and objective measurement of coverage.

3. **Empirical findings:** A systematic study demonstrating that even frontier reasoning models exhibit *pronounced mode collapse*—high Validity coupled with degrading Uniqueness and Recovery as $|\mathcal{H}_O|$ increases.

4. **Methodological contribution:** A reusable framework for analyzing hypothesis-generation capabilities, designed as a controlled probe for developing improved sampling strategies rather than a competitive benchmark.

We emphasize that HypoSpace does not claim to evaluate real-world scientific discovery. Rather, it abstracts core ingredients of set-valued inference—consistency checking and combinatorial hypothesis spaces—into controlled settings where ground truth is fully specified. This design choice prioritizes measurement precision and cross-model

comparability.

## 2. Related Work

Scientific-discovery benchmarks probe LLM reasoning in domains such as equation discovery and physics-inspired tasks, and in end-to-end AI-scientist loops (Shojaee et al., 2024; 2025; Koblischke et al., 2025; Coignion et al., 2024; Chen et al., 2025), but typically optimize for single-answer correctness and assume a single ground truth per input. Creativity-aware evaluations, rooted in novelty/appropriateness and the fluency/originality/flexibility triad (Guilford, 1950; Amabile, 1996; Boden, 2004), have begun to incorporate diversity (e.g., HypoBench, IdeaBench, CreativEval, LiveIdeaBench) (Liu et al., 2025; Guo et al., 2025b; DeLorenzo et al., 2024; Ruan et al., 2024), and propose open-ended/axiomatic frameworks for originality and distributional creativity (Nagarajan et al., 2025; Wang et al., 2024a). Unlike these, we evaluate set-valued inference with deterministic validators and exactly enumerated admissible sets, enabling precise, model-agnostic measurement of Validity, Uniqueness, and Recovery without LLM-as-judge. (Extended related work in Appendix B.)

## 3. Diagnostics for Creative Hypotheses Generation : Details of Formulation and Indicators

**Problem setup.** Let $\mathcal{O}$ denote the full observation space and let $O \subseteq \mathcal{O}$ be the subset revealed for a given instance. Let $\mathcal{H}$ be the overall hypothesis space and $\mathcal{H}_O \subseteq \mathcal{H}$ the subset of hypotheses that are consistent with $O$. Our diagnostic suite assumes that, for each instance, $\mathcal{H}_O$ is *explicitly enumerated*, enabling exact validity checks and direct measurement of coverage without LLM-as-judge circularity. We require three properties: (i) **soundness** (every $h \in \mathcal{H}_O$ is consistent with $O$), (ii) **completeness** (no valid hypothesis is omitted), and (iii) **controllability** (the size of the ground-truth admissible set $|\mathcal{H}_O|$ is controlled by task parameters—e.g., node count, grid dimensions, operator set/depth).

**Sampling protocol.** We evaluate models as *samplers over hypothesis sets*. From a fixed prompt and decoding setup, we draw $N$ independent samples $P = \{\tilde{h}_1, \ldots, \tilde{h}_N\}$, logging both the proposals and their order for novelty checks. In most experiments we set $N = |\mathcal{H}_O|$ to facilitate coverage analyses, but other sampling budgets are possible.

### 3.1. Indicators

We summarize behavior along three indicators that disentangle selection fidelity, non-redundancy, and coverage. Throughout, $\mathrm{val}_O(\tilde{h}) \in \{0, 1\}$ denotes the task-specific validity check and $\mathrm{nov}(\tilde{h}; A_{\tilde{h}}) \in \{0, 1\}$ indicates whether $\tilde{h}$

is distinct relative to the set of earlier samples $A_{\tilde{h}} \subseteq P$ (order-respecting).

**Validity (VR; appropriateness).**

$$\mathrm{VR}(P) \;=\; \frac{1}{N} \left| \{ \, \tilde{h} \in P \mid \mathrm{val}_O(\tilde{h}) = 1 \, \} \right|. \quad (1)$$

VR measures selection fidelity: the share of proposals that satisfy all observations.

**Novelty/Uniqueness (NR; originality).**

$$\mathrm{NR}(P) \;=\; \frac{1}{N} \left| \{ \, \tilde{h} \in P \mid \mathrm{nov}(\tilde{h}; A_{\tilde{h}}) = 1 \, \} \right|. \quad (2)$$

NR quantifies non-redundancy within a sampling run.

**Recovery (RR; fluency/coverage).**

$$\mathrm{RR}(P) = \frac{1}{|\mathcal{H}_O|} \left| \{ \tilde{h} \in P \mid \mathrm{val}_O(\tilde{h}) = 1 \right.$$
$$\left. \wedge \; \mathrm{nov}(\tilde{h}; A_{\tilde{h}}) = 1 \} \right|. \quad (3)$$

RR measures coverage of the *enumerated* valid set and integrates validity and non-redundancy.

**Distinctness criteria.** Novelty/uniqueness is task-specific: labeled-edge equality (causal DAGs), voxelwise tensor equality (3D reconstructions), and a mechanistic canonicalizer for local Boolean equivalences (genetic interaction).

### 3.2. Intended use and scope

Our suite is a *diagnostic*, not a leaderboard: it isolates underdetermination, supports controlled ablations (sampling budget, model class), and reports interpretable indicators and coverage curves rather than a single scalar score. It does not claim real-world scientific discovery; instead, it provides a calibrated probe for set-valued inference.

## 4. Why Coverage Collapse Persists Under Peaked Generators

This section provides a simple analysis of why *coverage* (Recovery) can remain low even when Validity is high. The key point is not that full recovery is mathematically impossible, but that for *peaked* hypothesis distributions, the sampling budget required to cover the admissible set can be astronomically large (and may be infinite if some admissible hypotheses have zero probability).

**Setup.** Let $\mathcal{H}_O = \{h_1, \ldots, h_M\}$ denote the finite admissible hypothesis set for an instance (enumerable in HypoSpace). Consider a model-induced distribution $p$ over $\mathcal{H}_O$, where $p(h) \geq 0$ and $\sum_{h \in \mathcal{H}_O} p(h) = 1$. We draw $N$ i.i.d. samples $h^{(1)}, \ldots, h^{(N)} \sim p$. Define the (unnormalized) coverage count as

$$C_N \;=\; \left| \{ h \in \mathcal{H}_O : \exists t \in \{1, \ldots, N\} \text{ s.t. } h^{(t)} = h \} \right|,$$

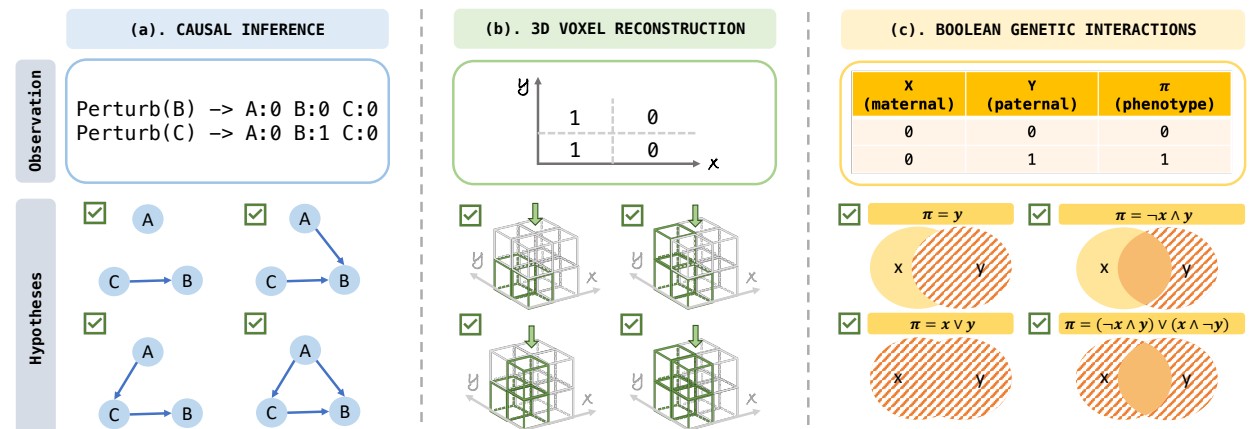

*Figure 2.* **HypoSpace task instantiations.** (a) *Causal inference from perturbations*: Given intervention observations (e.g., perturbing node C affects nodes B), models propose causal DAGs consistent with the data. Multiple valid graph structures can explain the same perturbation patterns. (b) *3D voxel reconstruction under gravity*: From top-down 2D projections, models reconstruct 3D voxel structures satisfying projection constraints and gravity (stacking rules). The same projection admits multiple valid 3D configurations. (c) *Boolean genetic interactions*: From phenotype observations of parental combinations, models propose Boolean expressions relating inputs to outputs. Expressions that collapse to the same form under our restricted canonicalizer count as one hypothesis. Each domain exemplifies scientific underdetermination where multiple mechanistically distinct hypotheses explain identical observations, enabling measurement of Validity, Uniqueness, and Recovery across enumerated hypothesis spaces.

and the Recovery rate as $\mathrm{RR}(N) = C_N/M$.

**Expected Recovery.** For any fixed $h \in \mathcal{H}_O$, the probability that $h$ is never sampled in $N$ draws is $(1 - p(h))^N$, hence the probability that it is observed at least once is $1 - (1 - p(h))^N$. By linearity of expectation,

$$\mathbb{E}[C_N] = \sum_{h \in \mathcal{H}_O} \Big(1 - (1 - p(h))^N\Big),$$
$$\mathbb{E}[\mathrm{RR}(N)] = \frac{1}{M} \sum_{h \in \mathcal{H}_O} \Big(1 - (1 - p(h))^N\Big). \tag{4}$$

**Asymptotics versus practical budgets.** If every admissible hypothesis has nonzero probability mass, i.e., $\min_{h \in \mathcal{H}_O} p(h) > 0$, then $\mathbb{P}(C_N = M) \to 1$ as $N \to \infty$ (equivalently, $\mathbb{E}[\mathrm{RR}(N)] \to 1$). However, the *rate* of convergence depends critically on the smallest tail probabilities. To see this, for a hypothesis with $p(h) = \varepsilon \ll 1$,

$$1 - (1 - \varepsilon)^N \approx N\varepsilon \quad \text{when } N\varepsilon \ll 1, \tag{5}$$

so achieving a constant probability of observing $h$ requires $N = \Theta(1/\varepsilon)$. Thus, if a large fraction of $\mathcal{H}_O$ lies in the tail with exponentially small probabilities (a common pattern under peaked generators), the sampling budget required to achieve substantial Recovery can be exponential in the problem size.

**Peaked distributions imply slow coverage.** A convenient way to formalize "peakedness" is via a head–tail decomposition: let $S \subset \mathcal{H}_O$ be a small "head" set of size $|S| = K \ll M$ that carries most of the probability mass, and define $\alpha = \sum_{h \in S} p(h)$ (typically $\alpha \approx 1$). Then combining (4) with the bound $1 - (1 - x)^N \leq \min\{1, Nx\}$ yields

$$\mathbb{E}[C_N] = \sum_{h \in S} \Big(1 - (1 - p(h))^N\Big)$$
$$+ \sum_{h \notin S} \Big(1 - (1 - p(h))^N\Big)$$
$$\leq K + N \sum_{h \notin S} p(h) = K + N(1 - \alpha). \tag{6}$$

When $\alpha$ is close to 1 (most mass concentrated on a few modes), the expected number of distinct hypotheses discovered grows at most as $K + N(1 - \alpha)$, i.e., with a very small slope determined by the tail mass. Normalizing by $M$ shows that Recovery can remain small even for large $N$ whenever $M$ is large and $1 - \alpha$ is tiny.

**Why high Validity can coexist with low Recovery.** If the model's support is contained in $\mathcal{H}_O$ (or nearly so), then Validity can be close to 1 even when $p$ is highly peaked. In that case, repeated sampling quickly re-discovers the same few admissible hypotheses (high Validity) while failing to cover the long tail (low Recovery). This yields the characteristic "high-VR, low-RR" regime observed in our experiments.

**Zero-probability hypotheses.** The above discussion assumes $p(h) > 0$ for all $h \in \mathcal{H}_O$. In practice, some admissible hypotheses may be effectively unreachable under a given prompting/decoding scheme, i.e., $p(h) = 0$ for some

valid $h$. In this case, full Recovery is impossible regardless of sampling budget, and $\mathrm{RR}(N)$ is bounded away from 1 for all $N$.

**Takeaway and Implications for LLMs.** Our analysis is *conditional*: coverage collapse is not a property of LLMs per se, but of any generator inducing a sufficiently peaked distribution over the admissible set. In principle, a generator assigning near-uniform mass (or actively reweights away from previously generated hypotheses) could achieve efficient coverage. However, modern LLMs—trained with likelihood- and preference-based objectives and decoded via high-probability continuations—empirically produce highly concentrated distributions. In all domains we study, frontier models exhibit this peaked behavior, and mitigating coverage collapse requires reshaping the sampling distribution rather than simply drawing more samples.

## 5. HypoSpace Construction

We instantiate three structured diagnostics (Figure 2) that mirror common patterns of scientific inference while allowing exact enumeration of $\mathcal{H}_O$: causal graphs from perturbations, gravity-constrained 3D voxel worlds from top-down projections, and Boolean genetic interactions from phenotype observations.

### 5.1. Causal inference from perturbations

**Instance.** As shown in Figure 2a, let $G^\star = (V, E^\star)$ be a latent DAG over $n$ labeled nodes. We observe single-node interventions: perturbing node $v_i$ affects exactly its descendants in $G^\star$, producing a binary response vector $x^{(k)} \in \{0,1\}^n$ for each intervention $s^{(k)} \in V$. The observation set is $O = \{(s^{(k)}, x^{(k)})\}_{k=1}^m$.

**Generation.** At each step, the model proposes a candidate DAG $\tilde{G} \leftarrow F_{\mathrm{LLM}}(O, A_{\tilde{G}}, T_{\mathrm{prompt}})$, where $A_{\tilde{G}}$ is the history.

**Validity and distinctness.** A proposal is valid iff it reproduces all observed effects: $\mathrm{val}_O(\tilde{G}) = 1 \Leftrightarrow \Phi_{\tilde{G}}(s^{(k)}) = x^{(k)} \ \forall k$. Distinctness is on labeled nodes: two DAGs are identical if their labeled edge sets match; equivalently $\mathrm{Canon}(G) = \mathrm{Canon}(G')$.

**Scoring and difficulty.** Given $N$ samples $P = (\tilde{G}_i)_{i=1}^N$, we report VR/NR/RR as in Section 3. Difficulty is controlled by $n$ and $m$ (more nodes/interventions typically enlarge $|\mathcal{H}_O|$).

### 5.2. 3D understanding under gravity

**Instance.** As indicated in Figure 2b, an observation is a binary top-down projection $V \in \{0,1\}^{M \times M}$ where $V_{i,j} = 1$ indicates at least one occupied voxel in column $(i, j)$. Hypotheses are voxel stacks $\mathcal{H} = \mathcal{H}(M, K) \subseteq \{0,1\}^{K \times M \times M}$ with $K$ discrete layers (layer 1 bottom).

**Generation.** At each step, the model proposes $\tilde{h} \in \mathcal{H} \leftarrow F_{\mathrm{LLM}}(V, A_{\tilde{h}}, T_{\mathrm{prompt}}; M, K)$.

**Validity and distinctness.** A reconstruction is valid iff

$$\mathrm{val}_V(\tilde{h}) = 1 \iff \left[ \forall i, j : \bigvee_{k=1}^K \tilde{h}_{k,i,j} = V_{i,j} \right] \\ \wedge \left[ \forall i, j, k > 1 : \tilde{h}_{k,i,j} \le \tilde{h}_{k-1,i,j} \right], \tag{7}$$

enforcing projection consistency and bottom-contiguous gravity. Distinctness is voxelwise equality.

**Scoring and difficulty.** We report VR/NR/RR as in Section 3. Difficulty is controlled by grid size $M$, height budget $K$, and projection density; the number of valid completions grows rapidly with these parameters.

### 5.3. DNA interaction via Boolean programs

**Instance.** As in Figure 2 (c), inputs are maternal/paternal phenotypes $x, y \in \{0,1\}$ and output $\pi \in \{0,1\}$. An instance specifies an operator set $\mathcal{F}$ (e.g., $\{\neg, \wedge, \vee\}$) and a depth bound $d$ defining the hypothesis space $\mathcal{H} = \mathcal{H}(\mathcal{F}, d)$ of expression trees over $\{x, y\}$ (optionally constants). Each $h \in \mathcal{H}$ induces $f_h : \{0,1\}^2 \to \{0,1\}$. Observations are pairs $O = \{((x^{(i)}, y^{(i)}), \pi^{(i)})\}_{i=1}^m$.

**Generation.** The model proposes $\tilde{h} \in \mathcal{H} \leftarrow F_{\mathrm{LLM}}(O, A_{\tilde{h}}, T_{\mathrm{prompt}}; \mathcal{F}, d)$.

**Validity and distinctness.** Validity requires functional agreement on observed pairs: $\mathrm{val}_O(\tilde{h}) = 1 \Leftrightarrow f_{\tilde{h}}(x^{(i)}, y^{(i)}) = \pi^{(i)} \ \forall i$. Distinctness is defined by a mechanistic canonicalizer that collapses only local algebraic symmetries implemented in our code: commutativity, idempotence, and associativity flattening for repeated identical operators; two expressions are identical iff $\mathrm{Canon}(h) = \mathrm{Canon}(h')$.

**Scoring and difficulty.** We report VR/NR/RR as in Section 3. Difficulty is controlled by the operator set $\mathcal{F}$, depth $d$, and the number/coverage of observation pairs; these knobs govern $|\mathcal{H}_O|$ and the combinatorics of valid programs.

## 6. Experiments

We evaluate a mix of instruction-tuned and "thinking" LLMs on three diagnostic tasks with enumerated valid sets: Causal Inference, 3D Understanding, and Boolean Genetic Interaction. The model suite includes GPT-4o (Hurst et al., 2024), GPT-5 (OpenAI, 2025), Gemini-2.5-Pro (Comanici et al., 2025), Claude-Opus-4 (Anthropic, 2025), DeepSeek-R1 (Guo et al., 2025a), LLaMA-3.3-70B-Instruct (Dubey et al., 2024), and Grok-4 (xAI, 2025). All models were accessed via OpenRouter (OpenRouter, 2025), except GPT-5, which was evaluated via the OpenAI API. Provider/version

*Table 1.* **HypoSpace evaluation on Causal Inference.** We report Validity Rate (VR; fraction of proposed hypotheses consistent with observations), Novelty/Uniqueness Rate (NR; non-redundancy among proposals), and Recovery Rate (RR; coverage of the enumerated admissible set $\mathcal{H}_O$). Difficulty increases from Level 1 to 3 as the hypothesis space size $|H_O|$ grows. Numbers are mean $\pm$ standard deviation across instances; **Bold** indicates the best result in each row; underline indicates the second-best result.

| Difficulty | Metric | Reasoning Models | | | | | Non-Reasoning Models | |
| --- | --- | --- | --- | --- | --- | --- | --- | --- |
| | | **GPT-5** | **Gemini-2.5-Pro** | **Claude-Opus-4** | **Deepseek-R1** | **Grok-4** | **GPT-4o** | **LLaMa-3.3-70b-Instruct** |
| 1 (nodes=4) | VR | **100.00% ± 0.00%** | **100.00% ± 0.00%** | 89.30% ± 21.70% | **100.00% ± 0.00%** | **100.00% ± 0.00%** | 73.80% ± 32.70% | 30.00% ± 40.10% |
| | NR | **100.00% ± 0.00%** | **100.00% ± 0.00%** | 86.20% ± 20.80% | 98.20% ± 4.50% | **100.00% ± 0.00%** | 83.20% ± 23.00% | 83.30% ± 27.00% |
| | RR | **100.00% ± 0.00%** | **100.00% ± 0.00%** | 77.50% ± 27.10% | 98.20% ± 4.50% | **100.00% ± 0.00%** | 60.90% ± 34.50% | 24.00% ± 34.40% |
| 2 (nodes=5) | VR | **100.00% ± 0.00%** | **100.00% ± 0.00%** | 80.10% ± 23.10% | 99.00% ± 3.30% | **100.00% ± 0.00%** | 49.00% ± 36.20% | 17.60% ± 28.10% |
| | NR | **100.00% ± 0.00%** | **100.00% ± 0.00%** | 79.90% ± 23.60% | 93.50% ± 12.00% | **100.00% ± 0.00%** | 83.50% ± 24.60% | 86.00% ± 21.00% |
| | RR | **100.00% ± 0.00%** | **100.00% ± 0.00%** | 65.40% ± 28.40% | 92.50% ± 12.80% | **100.00% ± 0.00%** | 38.30% ± 32.80% | 17.60% ± 28.10% |
| 3 (nodes=6) | VR | **100.00% ± 0.00%** | 99.80% ± 0.80% | 59.30% ± 21.20% | 98.20% ± 3.00% | **100.00% ± 0.00%** | 72.80% ± 34.10% | 10.40% ± 26.00% |
| | NR | 99.20% ± 1.40% | 99.40 % ± 1.30% | 90.00% ± 7.90% | 81.20% ± 9.80% | **99.80% ± 1.20%** | 22.90% ± 27.20% | 38.00% ± 22.40% |
| | RR | 99.20% ± 1.40% | 99.20% ± 1.40% | 51.10% ± 22.10% | 79.50% ± 9.30% | **99.80% ± 1.20%** | 6.70% ± 8.30% | 2.30% ± 2.70% |

*Table 2.* **HypoSpace evaluation on 3D Understanding.** We report VR, NR, and RR under three difficulty levels with increasing $|H_O|$. Results are mean $\pm$ standard deviation across instances; **Bold** marks the best result; underline marks the second-best.

| Difficulty | Metric | Reasoning Models | | | | | Non-Reasoning Models | |
| --- | --- | --- | --- | --- | --- | --- | --- | --- |
| | | **GPT-5** | **Gemini-2.5-Pro** | **Claude-Opus-4** | **Deepseek-R1** | **Grok4** | **GPT-4o** | **LLaMa-3.3-70b-Instruct** |
| 1 (tp=1) | VR | **100.00% ± 0.00%** | **100.00% ± 0.00%** | **100.00% ± 0.00%** | 96.30% ± 10.50% | **100.00% ± 0.00%** | 63.00% ± 24.60% | 18.50% ± 16.60% |
| | NR | **100.00% ± 0.00%** | **100.00% ± 0.00%** | **100.00% ± 0.00%** | **100.00% ± 0.00%** | **100.00% ± 0.00%** | 96.30% ± 10.50% | 88.90% ± 15.70% |
| | RR | **100.00% ± 0.00%** | **100.00% ± 0.00%** | **100.00% ± 0.00%** | 96.30% ± 10.50% | **100.00% ± 0.00%** | 59.30% ± 26.20% | 18.50% ± 16.60% |
| 2 (tp=2) | VR | **100.00% ± 0.00%** | **100.00% ± 0.00%** | 90.40% ± 12.40% | 99.30% ± 2.80% | **100.00% ± 0.00%** | 38.50% ± 19.20% | 10.40% ± 16.70% |
| | NR | **100.00% ± 0.00%** | **100.00% ± 0.00%** | 87.80% ± 12.30% | 99.60% ± 2.00% | **100.00% ± 0.00%** | 75.20% ± 14.80% | 84.40% ± 13.30% |
| | RR | **100.00% ± 0.00%** | **100.00% ± 0.00%** | 78.10% ± 11.30% | 98.90% ± 3.30% | **100.00% ± 0.00%** | 30.70% ± 12.70% | 8.90% ± 13.60% |
| 3 (tp=3) | VR | **100.00% ± 0.00%** | 99.90% ± 0.70% | 62.30% ± 19.70% | 99.00% ± 1.60% | 99.90% ± 0.70% | 19.40% ± 9.30% | 10.50% ± 13.10% |
| | NR | 98.80% ± 2.20% | 95.20% ± 4.60% | 81.10% ± 10.60% | 86.20% ± 5.20% | **99.90% ± 0.70%** | 57.90% ± 14.00% | 66.70% ± 14.30% |
| | RR | 98.80% ± 2.20% | 95.10% ± 4.80% | 48.70% ± 11.50% | 85.20% ± 4.90% | **99.80% ± 0.90%** | 14.30% ± 6.50% | 6.00% ± 5.60% |

metadata and full prompts/decoding settings are deferred to the Appendix C. Our total API budget was $\sim$ \$1,000.

We group models into **reasoning** (a.k.a. "thinking") and **non-reasoning** (instruction-tuned) categories. Reasoning models, by default, produce explicit intermediate rationales and are marketed as reasoning-optimized; non-reasoning models typically return short direct answers unless prompted otherwise. We score only the final structured hypothesis, not the rationale text.

### 6.1. Experimental details and difficulty

Each task exposes natural knobs that scale the size of the admissible set $|\mathcal{H}_O|$. **Causal Inference** varies the number of nodes and observed interventions; **3D Voxel Reconstruction** varies the number of top-down views at fixed height budget; **Boolean Genetic Interaction** varies the Boolean operator set/depth and observation coverage. We use three regimes (simple/medium/hard). For transparency, the instance settings are printed in the Tables 1-3. Unless noted otherwise, for each instance we draw $N$ independent sam-

ples with $N = |\mathcal{H}_O|$, compute VR/NR/RR on the $N$ proposals, and aggregate by averaging across instances within task/difficulty (reporting mean±std across repeated runs).

**Validation and distinctness.** Outputs follow strict schemas and are checked by deterministic validators (forward simulation for causal DAGs; projection and gravity for 3D; functional agreement on observed pairs for DNA). Distinctness is assessed via labeled-edge equality (causal), tensor equality (3D), and a mechanistic canonicalizer that collapses local algebraic symmetries (DNA).

### 6.2. Results by task

Table 1-Table 3 summarizes Validity (VR), Uniqueness (NR), and Recovery (RR) across difficulty for the three diagnostics. In causal inference, simple/medium regimes are near-ceiling for several reasoning models; in the hard case (6 nodes) the top reasoning models sustain near $100\%$ VR and high NR/RR, while other reasoning models show small but consistent NR/RR gaps and non-reasoning models lag in VR (hence RR). In 3D voxel reconstruction, most

*Table 3.* **HypoSpace evaluation on DNA Interaction.** We evaluate models using VR, NR, and RR as difficulty increases (Level 1–3) with larger admissible sets $|H_O|$. Entries show mean $\pm$ standard deviation across instances; **Bold** marks the best result; underline marks the second-best..

| Difficulty | Metric | Reasoning Models | | | | | Non-Reasoning Models | |
| | | GPT-5 | Gemini-2.5-Pro | Claude-Opus-4 | Deepseek-R1 | Grok4 | GPT-4o | LLaMa-3.3-70b-Instruct |
|---|---|---|---|---|---|---|---|---|
| | VR | **100.00% ± 0.00%** | **100.00% ± 0.00%** | 88.30% ± 16.50% | 83.90% ± 15.50% | 89.30% ± 21.70% | 88.40% ± 26.20% | 95.60% ± 18.70% |
| 1 (basic) | NR | **100.00% ± 0.00%** | **100.00% ± 0.00%** | 69.90% ± 19.40% | 82.90% ± 16.30% | 89.30% ± 21.70% | 38.10% ± 21.40% | 30.10% ± 14.70% |
| | RR | **100.00% ± 0.00%** | **100.00% ± 0.00%** | 69.90% ± 19.40% | 82.90% ± 16.30% | 89.30% ± 21.70% | 38.10% ± 21.40% | 29.00% ± 15.70% |
| | VR | 74.90% ± 21.40% | 72.30% ± 21.10% | 52.70% ± 23.20% | 57.20% ± 21.70% | 52.20% ± 20.40% | **83.00% ± 27.50%** | 55.80% ± 45.50% |
| 2 (extended) | NR | **74.90% ± 21.40%** | 72.30% ± 21.10% | 54.90% ± 24.50% | 57.20% ± 21.70% | 52.20% ± 20.40% | 31.60% ± 21.50% | 18.40% ± 16.70% |
| | RR | **72.30% ± 21.30%** | 72.30% ± 21.10% | 48.40% ± 23.20% | 53.90% ± 24.50% | 50.80% ± 20.70% | 27.30% ± 22.90% | 14.90% ± 17.80% |
| | VR | 65.10% ± 12.50% | 52.20% ± 15.50% | 36.90% ± 15.90% | 41.00% ± 13.60% | 40.60% ± 15.00% | **68.10% ± 37.40%** | 66.60% ± 43.90% |
| 3 (full) | NR | **49.90% ± 14.00%** | 48.90% ± 14.90% | 24.20% ± 10.20% | 36.60% ± 12.30% | 37.90% ± 15.20% | 21.10% ± 14.30% | 14.30% ± 9.90% |
| | RR | **48.00% ± 14.40%** | 47.40% ± 13.40% | 23.80% ± 10.20% | 36.00% ± 12.80% | 36.50% ± 14.90% | 14.10% ± 10.50% | 11.00% ± 9.40% |

models attain high VR with 1–2 views, but at 3 views ($|\mathcal{H}_O|$=27) gaps widen: frontier reasoning models preserve NR/RR whereas others repeat proposals early, depressing RR. Boolean genetic interactions is most discriminative: as operator set/depth and observation coverage grow, NR and RR drop markedly for all models even when VR remains moderate; the canonicalizer collapses superficial variants, revealing limited exploration of distinct mechanisms.

Because the strongest reasoning models remain near ceiling on the original causal and 3D scales, we further evaluate GPT-5, Gemini-2.5-Pro, and Grok-4 on larger admissible sets: $|\mathcal{H}_O| = 160$ for causal inference and $|\mathcal{H}_O| = 125$ for 3D voxel reconstruction. As shown in Table 4, the validity–coverage gap becomes clearer at these larger scales. All three models retain $100\%$ VR, but their NR and RR drop substantially, indicating that high validity does not necessarily imply broad coverage once the admissible hypothesis space grows.

Additionally, in several settings, NR can exceed VR because NR counts unique proposals regardless of validity, whereas VR counts valid proposals. Thus, NR > VR indicates diverse but often invalid outputs. This distinction motivates RR, which measures valid and unique discoveries jointly and separates conservative repetition from diversity without constraint satisfaction.

### 6.3. Cross-task observations

Across all three domains we see the consistent trend: as $|\mathcal{H}_O|$ grows, Uniqueness and Recovery drop even while frontier reasoning models (GPT-5, Gemini-2.5-Pro, Claude-Opus-4, DeepSeek-R1, Grok-4) remain higher on Validity. By task, Causal Inference is most tractable at our scales (several models near ceiling), 3D Voxel is intermediate with collapse emerging in the hardest multi-view settings, and Boolean Genetic Interactions is most discriminative—its large program space, even after canonicalization, yields

the sharpest coverage deficits. Reasoning-capable models consistently beat non-reasoning baselines on NR/RR at medium–hard difficulties, indicating that explicit reasoning mitigates—but does not eliminate—mode collapse.

### 6.4. Where coverage fails: output-level failure modes

Declining RR indicates reduced coverage of the admissible hypothesis set, but it does not by itself identify a single failure mechanism. We therefore inspect the generated outputs to separate several sources of coverage loss. In settings where models maintain high VR but RR drops, the deficit is primarily due to repeated generation of already recovered valid hypotheses, which is consistent with mode collapse. This pattern is especially clear in the larger-scale causal and 3D settings, where frontier reasoning models retain $100\%$ VR while NR/RR decrease substantially.

In harder symbolic settings, however, coverage loss is more heterogeneous. On Boolean (Hard), weaker and non-reasoning models often fail through exact duplication, whereas stronger reasoning models exhibit a mixture of constraint violations, invalid hypotheses, and canonical duplicates. Thus, mode collapse is an important failure mode, but not the only one. More generally, reduced RR should be interpreted as a coverage diagnostic whose underlying cause can vary across tasks and models. This motivates reporting VR, NR, and RR jointly, together with output-level analyses of duplication, invalidity, and constraint violations. A detailed failure-mode analysis on Boolean (Hard) is provided in Appendix G.

### 6.5. A diagnostic intervention: complexity-stratified decoding

Our analysis in Appendix D shows that LLMs exhibit a strong *simplicity bias*: they preferentially generate low-complexity hypotheses and rarely explore more complex

*Table 4.* **Larger-scale experiments.** Parentheses show percentage-point drops relative to the hardest original settings in Tables 1 and 2.

| | Causal ($|\mathcal{H}_O| = 160$) | | | 3D ($|\mathcal{H}_O| = 125$) | | |
|---|---|---|---|---|---|---|
| Model | VR | NR | RR | VR | NR | RR |
| GPT-5 | 100.0% (0.0 ↓) | 71.2% (28.0 ↓) | 71.2% (28.0 ↓) | 100.0% (0.0 ↓) | 75.2% (23.6 ↓) | 75.2% (23.6 ↓) |
| Gemini-2.5-Pro | 100.0% (0.0 ↓) | 72.4% (27.0 ↓) | 72.4% (26.8 ↓) | 100.0% (0.0 ↓) | 64.8% (30.4 ↓) | 64.8% (30.3 ↓) |
| Grok-4 | 100.0% (0.0 ↓) | 57.7% (42.1 ↓) | 57.7% (42.1 ↓) | 100.0% (0.0 ↓) | 79.2% (20.7 ↓) | 79.2% (20.6 ↓) |

*Table 5.* **Baseline vs. stratified decoding (Boolean genetic interactions).** We report baseline and Δ (Strat. − Base), in %.

| | RR | | Simple ($c \leq 3$) | | Complex ($c > 3$) | |
|---|---|---|---|---|---|---|
| Model | Base | Δ | Base | Δ | Base | Δ |
| GPT-5 | 38.5% | -7.7 | 61.2% | -14.3 | 0.0% | +3.4 |
| Gemini-2.5-Pro | 37.2% | -9.0 | 59.2% | -14.3 | 0.0% | +0.0 |
| Claude-Opus-4 | 17.9% | +7.7 | 28.6% | +10.2 | 0.0% | +3.4 |
| DeepSeek-R1 | 25.6% | -19.2 | 40.8% | -30.6 | 0.0% | +0.0 |
| Grok-4 | 33.3% | +3.9 | 53.1% | -4.1 | 0.0% | +17.2 |
| GPT-4o | 5.1% | +9.0 | 8.2% | +14.2 | 0.0% | +0.0 |
| LLaMA-3.3-70B | 1.3% | +3.8 | 2.0% | +4.1 | 0.0% | +3.4 |

regions of the admissible space, even when many valid hypotheses reside there. To counteract this bias, we introduce a simple, training-free baseline that explicitly *stratifies generation by structural complexity*.

For each task, we define a natural, task-specific notion of hypothesis complexity—distinct from the task *difficulty levels* in Section 5, which reflect the size of the admissible set $|\mathcal{H}_O|$. Here, complexity refers to the structural richness of individual hypotheses: number of edges in a causal graph, number of operators in a Boolean expression, or number of occupied voxels in a 3D configuration. Instead of sampling from the model's unconstrained distribution, we iterate over complexity levels $c = \{0, 1, \ldots, c_{\max}\}$ and prompt the model to generate hypotheses of exactly complexity $c$, while still obeying all validity constraints and avoiding previously generated hypotheses. The query budget is allocated across complexity levels proportionally to the ground-truth distribution at each level, ensuring adequate coverage of both simple and complex hypotheses.

This complexity-stratified decoding procedure converts the model's implicit, highly biased search into an explicit breadth-first exploration over the hypothesis space. While still relying entirely on the model for generation and validation, it encourages exploration of higher-complexity hypotheses that the model would otherwise neglect, providing a complementary, training-free baseline to memory-based reject-duplicate decoding.

Table 5 compares memory-based decoding with complexity-stratified decoding. Stratified decoding improves overall Recovery Rate for 4/7 models, with the largest gains for GPT-4o (+9.0%) and Claude-Opus-4 (+7.7%). More importantly,

it enables non-zero recovery on complex hypotheses for 4 models, overcoming the 0% Complex RR baseline; Grok-4 shows the largest breakthrough ($0\% \rightarrow 17.2\%$). These gains come with trade-offs: strong baseline models (GPT-5, Gemini) lose Simple RR as generation budget is shifted toward higher-complexity regions, and DeepSeek-R1 degrades substantially (RR: $25.6\% \rightarrow 6.4\%$), suggesting sensitivity to explicit complexity constraints. Overall, stratified decoding is a training-free way to encourage exploration of complex hypothesis regions, though its effectiveness varies across models.

### 6.6. Effect of decoding and sampling budget

We further test whether coverage collapse can be resolved by simple decoding changes or by sampling more hypotheses. On the Boolean Genetic Interaction task under the hard setting, we sweep temperature and top-$p$ across three representative models, and separately increase the sampling budget to $3.0 \times |\mathcal{H}_O|$. As shown in Table 6, the qualitative pattern remains unchanged: diversity-oriented decoding does not consistently improve Recovery, and NR/RR vary only modestly across standard temperature and top-$p$ settings. Increasing the budget also fails to reliably improve RR; in some cases, models generate more duplicates or lose validity instead.

These results suggest that the coverage deficits are not artifacts of a particular decoding hyperparameter or insufficient sample count. Rather, models exhibit a structural bias toward narrow regions of the admissible hypothesis space, so additional randomness or more samples often revisits preferred modes instead of covering missing valid alternatives. This supports the need for explicitly coverage-aware strategies, such as memory-based rejection, complexity-stratified decoding, ensembling, or diversity-promoting prompts.

### 6.7. Further analysis

Due to page constraints, we defer several analyses to the Appendix: (i) cost and token usage across LLMs (Appendix A); (ii) model preferences under underdetermination (Appendix D); (iii) reasoning-trace analysis on the Boolean task (Appendix E); (iv) information-theoretic analysis of exploration dynamics (Appendix F); (v) failure mode decomposition and coverage analysis (Appendix G).

*Table 6.* **Decoding/budget ablations on Boolean (Hard).** RR is reported as mean ± std. Budget entries show RR with changes from the default budget.

| Model | T=0.2 | T=0.7 | T=1.0 | p=0.5 | p=0.8 | p=1.0 | $3.0 \times |\mathcal{H}_O|$ |
|---|---|---|---|---|---|---|---|
| GPT-4o | 8.6±6.4 | 14.0±5.8 | 10.5±6.5 | 16.5±7.1 | 12.6±4.5 | 14.0±5.8 | 12.6±4.5 (−1.4) |
| Claude-Opus-4 | 23.0±9.8 | 17.1±5.7 | 24.9±10.7 | 20.6±9.8 | 21.3±6.8 | 17.1±5.7 | 25.6±12.4 (+1.8) |
| Gemini-2.5-Pro | 46.0±9.6 | 49.9±13.9 | 50.3±15.3 | 47.2±16.7 | 47.0±21.2 | 49.9±13.9 | 37.3±2.7 (−9.9) |

# 7. Real-World Alignment Study

To connect our framework to real-world science, we instantiate it on an anonymized yeast vesicle-trafficking module from Costanzo et al. (2016): six functionally related genes with 6 single-knockout (KO) and 7 unique double-KO viability outcomes. We binarize fitness into viable vs. lethal and enumerate all Boolean hypotheses consistent with the observed perturbations. As shown in Figure 3, the consistent set $|H^*|$ contracts as double-KO observations are added (shaded range over all size-k double-KO subsets; line shows the average), with a synthetic-lethal pair producing a disproportionately large reduction—mirroring the "key experiment" effect in real discovery.

Table 7 shows LLM performance on this instance: stronger models (GPT-5, Grok-4, DeepSeek-R1, Gemini-2.5-Pro) achieve perfect validity and recover up to 100% of the hypothesis set, whereas weaker models fail to produce any consistent hypothesis (VR=0%) despite generating diverse outputs (high NR)—confirming that our metrics capture complementary dimensions of scientific reasoning.

# 8. Practical Use of HypoSpace

HypoSpace is intended as a diagnostic layer for LLM-assisted discovery pipelines. Its metrics inform three practical decisions. First, they support model selection: high-VR/low-RR models may suffice when only a few plausible hypotheses are needed, whereas higher-RR models are preferable when broad exploration matters. Second, metric profiles suggest interventions: low VR motivates stronger constraint prompting or external validation; high VR with low NR motivates reject-duplicate sampling or stratified decoding; and persistent RR deficits suggest insufficient exploration, motivating ensembling or diversity-promoting prompts. Third, HypoSpace supports deployment risk assessment: low RR indicates a risk of missing admissible alternatives, which is important when overlooked mechanisms or causal explanations may affect downstream scientific decisions.

# 9. Conclusion

We introduced HypoSpace, a diagnostic suite for set-valued evaluation of LLMs under underdetermination, with ex-

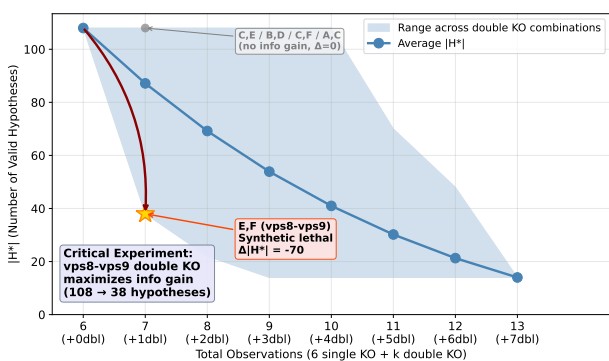

*Figure 3.* Using 6 single-KO and 7 unique double-KO observations from an anonymized vesicle-trafficking module, we enumerate the number of Boolean hypotheses consistent with the observations and show that $|H^*|$ shrinks as more double-KO measurements are added. The shaded band indicates the range across different choices of k double-KO observations, and highlights that the synthetic-lethal E–F double KO yields the largest contraction.

*Table 7.* LLM performance on genetic interaction inference (VR/NR/RR, %). Abbrev.: DS=DeepSeek-R1, Gem=Gemini-2.5-Pro, Cl=Claude-Opus-4, LLaMA=Llama-3.3-70B.

| | GPT-5 | Grok-4 | DS | Gem | Cl | LLaMA | GPT-4o |
|---|---|---|---|---|---|---|---|
| VR | 100.0 | 100.0 | 100.0 | 100.0 | 0.0 | 0.0 | 0.0 |
| NR | 100.0 | 100.0 | 85.7 | 85.7 | 85.7 | 71.4 | 57.1 |
| RR | **100.0** | **100.0** | 85.7 | 85.7 | 0.0 | 0.0 | 0.0 |

act validators and enumerated admissible sets across three structured domains. Our theoretical analysis shows that peaked hypothesis distributions make coverage collapse inevitable under realistic sampling budgets, even when validity remains high. Empirically, frontier models confirm this prediction: they maintain high Validity but exhibit limited Uniqueness and Recovery as $|\mathcal{H}_O|$ grows, a pattern that persists on real-world genetic interaction data (Section 7). Because all metrics are measured against enumerated ground truth, these failures are quantitative, not anecdotal. We further showed that stratified decoding can partially mitigate this collapse, suggesting that reshaping the sampling distribution is a more promising direction than simply increasing sampling budget.

## Acknowledgments

This research is partly supported by the National Research Foundation, Singapore, under its NRF AI-for-Science (AI4S) Challenge Grant (Award NRF-AI4SCH-2025-0007). Any opinions, findings, conclusions, or recommendations expressed in this material are those of the author(s) and do not reflect the views of the National Research Foundation, Singapore.

## Impact Statement

This paper advances methods for evaluating and diagnosing LLM-based scientific assistants. The primary positive impact is enabling more reliable and transparent assessment of model behavior in scientific reasoning settings, which may help reduce ungrounded claims and improve the safety of downstream scientific applications. Potential risks include misuse of evaluation protocols to overstate scientific capability, benchmark overfitting, and misinterpretation of results as biological conclusions rather than model-behavior measurements. To mitigate these risks, we focus on controlled tasks with explicit validators, report limitations of the hypothesis language and observation coverage, and release evaluation code and data processing details to support reproducibility and appropriate use. We do not anticipate direct negative impacts on individuals or deployment in high-stakes decision-making from this work.

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

# Appendix

## A. Analysis of cost and tokens across LLMs

Table 8 compares spend and token usage across seven models under a fixed nine-run budget. A run denotes one execution of a model on a fixed (task, difficulty) condition, during which the model generates $N$ samples; in Table 1, each row corresponds to one such run.

From the table, llama-3 is the most cost-efficient by a wide margin ($0.05/M tokens), followed by claude-opus-4 and grok-4 at $1.00/M and deepseek-r1 at $1.83/M; gpt-4o ($3.69/M), gpt-5 ($8.10/M), and gemini-2 ($8.69/M) are the most expensive per token, reflecting higher unit prices and longer average reasoning outputs. Despite low unit prices, deepseek-r1 and grok-4 accrued higher total spend due to large per-run budgets (∼2.5M tokens/run), whereas llama-3 and claude-opus-4 remained inexpensive both per token and in total.

*Table 8.* Model Cost Efficiency Analysis

| Model | Total Cost ($) | Total Tokens | Avg Tokens/Run |
|---|---|---|---|
| claude-opus-4 | 3.51 | 3,511,372 | 390,152 |
| deepseek-r1 | 41.45 | 22,701,210 | 2,522,356 |
| gemini-2.5-pro | 145.88 | 16,789,228 | 1,865,469 |
| gpt-4o | 10.66 | 2,886,161 | 320,684 |
| gpt-5 | 93.25 | 11,510,695 | 1,278,966 |
| grok-4 | 22.21 | 22,214,056 | 2,468,228 |
| llama-3.3-70b | 0.16 | 3,060,741 | 340,082 |

## B. Extended Related Work

**Benchmarks for Scientific Discovery with LLMs.** Recent work explores LLMs for scientific reasoning and discovery via domain benchmarks and end-to-end "AI scientist" setups. Shojaee et al. introduce equation-discovery tasks across physics and chemistry (Shojaee et al., 2024; 2025), and Koblischke et al. propose Gravity for physics-inspired reasoning (Koblischke et al., 2025). Coignion et al. assess LLM code generation (Coignion et al., 2024), while Chen et al. develop Auto-Bench to evaluate full AI-scientist loops including query generation and experiment planning (Chen et al., 2025). These efforts provide valuable snapshots of scientific reasoning skills but predominantly emphasize single-answer correctness or task completion and typically assume one ground-truth solution per input.

**Creativity-Aware Benchmarks.** Creativity has long been framed in psychology as producing outputs that are both *novel* and *appropriate*, with divergent thinking operationalized via *fluency*, *originality*, and *flexibility* (Guilford, 1950; Amabile, 1996), and further conceptualized by Boden's taxonomy of combinational, exploratory, and transformational creativity (Boden, 2004). In computational creativity, formal criteria for evaluating novelty/value were articulated

*Table 9.* Models evaluated with provider and snapshot.

| Model | Provider | Version/Snapshot |
|---|---|---|
| GPT-5 | OpenAI | 2025-08-07 |
| Gemini-2.5-Pro | Google | 2025-06-17 |
| Claude-Opus-4 | Anthropic | 2025-05-22 |
| DeepSeek-R1 | DeepSeek | 2025-01-20 |
| Grok-4 | xAI | 2025-07-09 |
| GPT-4o | OpenAI | 2024-05-13 |
| LLaMA-3.3-70B-Instruct | Meta | 2024-12-06 |

by Ritchie (Ritchie, 2007), and exploration-centric search (e.g., novelty search) was proposed to overcome objective-myopia (Lehman & Stanley, 2011). Building on this lineage, recent LLM benchmarks have begun to incorporate diversity/novelty. HypoBench (Liu et al., 2025) and IdeaBench (Guo et al., 2025b) quantify hypothesis novelty via instruction prompts and post-hoc analysis; CreativEval (DeLorenzo et al., 2024) measures fluency, flexibility, originality, and elaboration in LLM-generated code; LiveIdeaBench (Ruan et al., 2024) evaluates divergent thinking across Guilford's creativity dimensions; Nagarajan et al. design open-ended algorithmic tasks explicitly targeting originality and diversity, arguing next-token prediction is myopic for creative leaps and showing seed conditioning, multi-token objectives, and diffusion can elicit more diverse outputs (Nagarajan et al., 2025); and Wang et al. formalize Relative and Statistical Creativity as distributional indistinguishability from human creators, yielding practical measures for prompt-conditioned autoregressive models (Wang et al., 2024a). Beyond core NLP settings, Bhat et al. propose a domain-specific evaluation framework for marketing creativity with LLMs, operationalizing novelty and appropriateness using task-grounded criteria (Bhat et al., 2025).

However, most such evaluations rely on LLM-as-judge or human raters and lack a formally enumerated hypothesis space, making coverage and non-redundancy difficult to measure precisely. Our work complements these efforts by providing deterministic validators and exactly enumerated admissible sets, enabling model-agnostic, set-level measurement of Validity, Uniqueness, and Recovery.

## C. Model Metadata and Prompts

Table 9 lists the models included in our evaluation and clarifies the access path. "Provider" refers to the model's origin organization (e.g., OpenAI, Google, Anthropic), not the API gateway; all models were accessed via OpenRouter except `GPT-5`, which was called through the OpenAI API. Full prompts and decoding settings are provided in our anonymous repository.

## D. Model Preferences

We additionally conduct a preference analysis to examine whether models exhibit systematic biases under underdeter-

*Table 10.* Model preferences across three tasks. Generation Rate reports the fraction of valid generations that are Simple (S) vs. Complex (C). $RR_S$/$RR_C$ are recovery rates restricted to simple/complex ground-truth subsets. RR Gap = $RR_S - RR_C$.

| Model | Gen. Rate (S/C) | $RR_S$ | $RR_C$ | RR Gap | Overall RR | NR |
|---|---|---|---|---|---|---|
| **Causal (n=6 nodes; S: $\leq 3$ edges    C: $> 3$ edges)** | | | | | | |
| GPT-4o | 93% / 7% | 11% | 5% | +6% | 7% | 23% |
| LLaMA-3.3-70b-Instruct | 91% / 9% | 4% | 1% | +3% | 2% | 38% |
| GPT-5 | 47% / 53% | 100% | 99% | +1% | 99% | 99% |
| Claude-Opus-4 | 67% / 33% | 71% | 36% | +35% | 51% | 90% |
| Grok-4 | 47% / 53% | 100% | 100% | +0% | 100% | 100% |
| Gemini-2.5-Pro | 47% / 53% | 100% | 98% | +2% | 99% | 99% |
| DeepSeek-R1 | 54% / 46% | 96% | 66% | +30% | 79% | 81% |
| **3D (dim $3\times3\times3$; S: ground bias $\geq 50\%$    C: $< 50\%$)** | | | | | | |
| GPT-4o | 92% / 8% | 13% | 2% | +11% | 14% | 58% |
| LLaMA-3.3-70b-Instruct | 92% / 8% | 5% | 0% | +5% | 6% | 67% |
| GPT-5 | 64% / 36% | 100% | 97% | +3% | 99% | 99% |
| Claude-Opus-4 | 83% / 17% | 54% | 20% | +35% | 49% | 81% |
| Grok-4 | 63% / 37% | 100% | 99% | +1% | 100% | 100% |
| Gemini-2.5-Pro | 67% / 33% | 100% | 87% | +13% | 95% | 95% |
| DeepSeek-R1 | 70% / 30% | 91% | 73% | +18% | 85% | 86% |
| **Boolean (Hard; S: $\leq 3$ ops    C: 4–5 ops)** | | | | | | |
| GPT-4o | 100% / 0% | 21% | 0% | +21% | 14% | 21% |
| LLaMA-3.3-70b-Instruct | 100% / 0% | 17% | 0% | +17% | 11% | 14% |
| GPT-5 | 98% / 2% | 68% | 0% | +68% | 48% | 50% |
| Claude-Opus-4 | 98% / 2% | 35% | 0% | +35% | 24% | 24% |
| Grok-4 | 96% / 4% | 52% | 0% | +52% | 36% | 38% |
| Gemini-2.5-Pro | 98% / 2% | 67% | 0% | +67% | 47% | 49% |
| DeepSeek-R1 | 97% / 3% | 53% | 0% | +53% | 36% | 37% |

mination (e.g., favoring simpler hypotheses), and how such biases affect their ability to recover the full set of hypotheses consistent with the observations.

**Complexity definitions.**    We define task-specific complexity based on structural features:

- **Boolean:** Simple ($\leq 3$ operators) vs. Complex (4–5 operators).

- **Causal:** Simple ($\leq 3$ edges) vs. Complex ($> 3$ edges).

- **3D:** Simple (ground bias $\geq 50\%$) vs. Complex (ground bias $< 50\%$),

where ground bias is defined as

$$\text{ground bias} = \frac{\#\text{blocks on the ground layer}}{\#\text{total blocks placed}}.$$

**Metrics.**    For each model, we compute:

- **Generation Rate (Simple/Complex):** the distribution of complexity among all *valid* hypotheses generated by the model (including duplicates and excluding formatting errors), indicating the model's exploration bias during generation.

- **Recovery Rate ($RR_{Simple}$/$RR_{Complex}$):** the fraction of ground-truth hypotheses recovered by the model within the simple and complex subsets, quantifying capability at each complexity level.

- **Recovery Gap (RR Gap):** $RR_{Simple} - RR_{Complex}$.

- **Overall RR and NR:** consistent with the main paper metrics.

**Findings.**    Table 10 shows three consistent phenomena across tasks: (i) Bias toward less complex hypotheses: many models over-generate low-complexity forms; (ii) Higher recovery on less-complex sets: $RR_{Simple}$ exceeds $RR_{Complex}$ for weaker and mid-tier models, yielding a positive recovery gap; (iii) Collapse on complex regions: in Boolean (Hard), $RR_{Complex}$ is 0% across models, indicating systematic avoidance of deeper programs even when validity remains high.

## E. Reasoning-Trace Analysis on the Boolean Task

We further conduct a follow-up analysis of chain-of-thought reasoning traces on the Boolean task, where all models exhibit catastrophic failure (0% recovery on complex expressions). We construct a severely underdetermined instance with a single observation that admits 40 ground-truth Boolean hypotheses, and prompt three representative

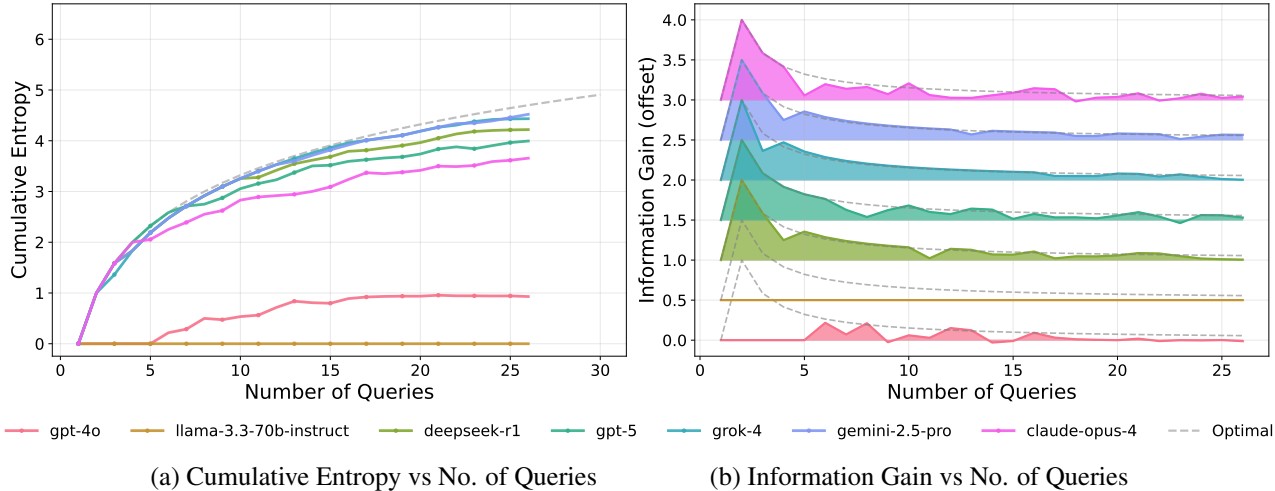

(a) Cumulative Entropy vs No. of Queries

(b) Information Gain vs No. of Queries

*Figure 4.* **Information-theoretic analysis of hypothesis space exploration in HypoSpace.** **(a)** Cumulative entropy as a function of sequential queries, measuring how uncertainty reduction varies across models as they sample from the hypothesis space. Steeper curves indicate more efficient exploration of distinct valid hypotheses. **(b)** Information gain per query (with 0.5 vertical offset for visual clarity). The curve shape reflects each model's marginal contribution of new information with additional samples, revealing differences in exploration strategies and susceptibility to mode collapse. Models with flatter curves show diminishing returns in hypothesis diversity, consistent with the Recovery Rate degradation observed in Tables 1-3.

models (GPT-4o, Grok-4, Gemini-2.5-Pro) to generate hypotheses with explicit explanations (*"Think step-by-step and explain your reasoning for each hypothesis you consider."*). We then quantify two behavioral dimensions using automated text processing.

**Automated trace metrics.**

- **Simplicity bias:** lexical pattern matching in the reasoning section, counting occurrences of cues such as `simple`, `basic`, `straightforward`, `most likely`, `enough to`, `should be sufficient`.

- **Termination pattern:** keyword detection to classify why the model stops generating:

  - **Sufficiency:** cues indicating subjective adequacy (e.g., `sufficient`, `should cover`, `thorough sample`, `unbounded`, `infinitely long`).
  - **Exhaustion:** cues claiming completeness (e.g., `all possible`, `exhausted`, `exhaustive`, `covered all`).

**Findings.** Table 11 shows that all three models explicitly prioritize low-complexity hypotheses, typically following a breadth-first exploration by operator count: they largely exhaust the 1–3 operator space while under-sampling the 4–5 operator region where most valid hypotheses reside in this instance. Moreover, the models exhibit distinct stopping behaviors yet converge on a shared failure mode of **false**

**convergence**—terminating generation based on subjective sufficiency or claimed exhaustion despite objective incompleteness. For example, GPT-4o claims systematic coverage with increasing complexity but recovers only 9/40 hypotheses; Grok-4 and Gemini-2.5-Pro cite practical sufficiency or unboundedness while still missing many hypotheses in an enumerable set.

*Table 11.* Reasoning-trace analysis on a severely underdetermined Boolean instance (1 observation; 40 ground-truth hypotheses). "Generated" and "Coverage" report the number of recovered hypotheses out of 40.

| Model | Simp. Bias | Termination | Generated | Coverage |
|---|---|---|---|---|
| Gemini-2.5-Pro | 3 | Sufficiency | 23/40 | 57.5% |
| GPT-4o | 3 | Exhaustion | 9/40 | 22.5% |
| Grok-4 | 3 | Sufficiency | 22/40 | 55.0% |

# F. Information-theoretic analysis of exploration dynamics

Although our diagnostics do not directly estimate entropies, they admit an information-theoretic view. Treat generation as a sequential process that grows a repertoire of mechanistic patterns. Let $\mathcal{H}_t$ be the multiset of hypotheses produced up to step $t$, $\mathcal{M}_t = M(\mathcal{H}_t)$ the induced set of patterns, and $p_t(m)$ the empirical frequency of pattern $m \in \mathcal{M}_t$. We define the stepwise information gain (entropy change) as

$$\Delta I_t = H_t - H_{t-1},$$
$$H_t := -\sum_{m \in \mathcal{M}_t} p_t(m) \log_2 p_t(m). \tag{8}$$

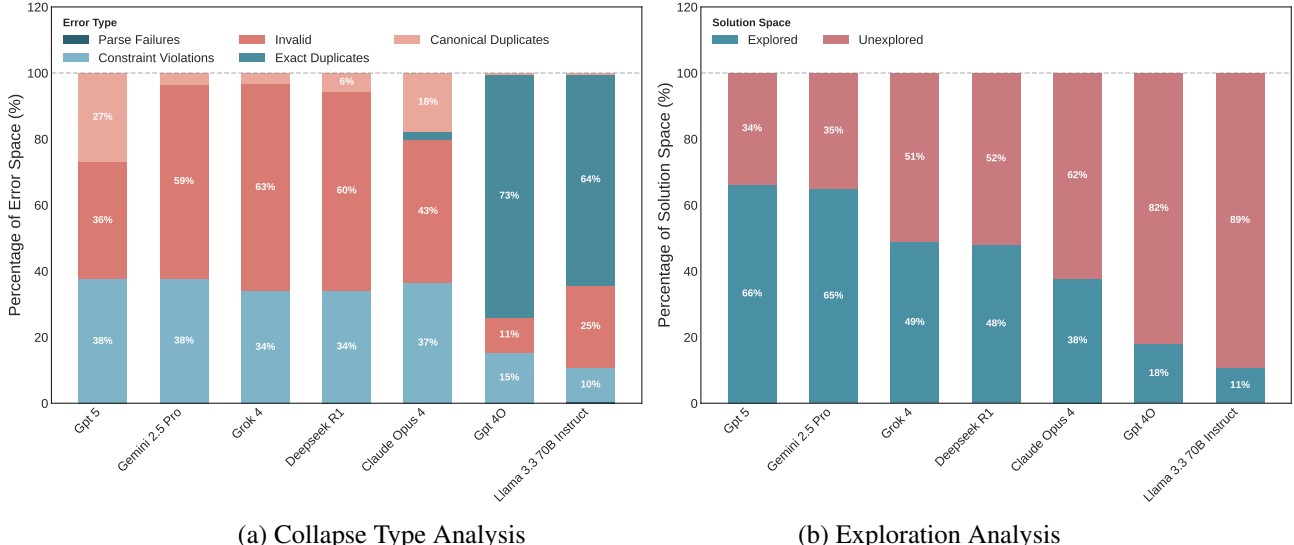

(a) Collapse Type Analysis        (b) Exploration Analysis

*Figure 5.* **Failure modes and exploration patterns in HypoSpace hypothesis generation.** **(a)** Error categorization across models: Distribution of failure types including parse failures (malformed outputs), constraint violations (structurally invalid hypotheses), invalid generations (inconsistent with observations), and duplicates (exact and canonical equivalences). Higher proportions of duplicates indicate mode collapse tendencies. **(b)** Hypothesis space coverage: Fraction of the enumerated admissible set $\mathcal{H}_O$ explored versus unexplored by each model. Limited exploration (lower blue bars) corresponds to reduced Recovery Rates and reveals the extent to which models fail to map the full space of valid explanations, even when maintaining high Validity.

Positive $\Delta I_t$ indicates expanding pattern diversity (exploration), while negative values signal convergence toward repeated patterns. Empirically, sustained positive $\Delta I_t$ aligns with higher novelty and improved Uniqueness/Recovery.

Figure 4 provides an information-theoretic lens on hypothesis space exploration, revealing fundamental differences in how models navigate the enumerated admissible sets. The cumulative entropy curves (Figure 4a) demonstrate that frontier reasoning models achieve steeper entropy growth, indicating more efficient discovery of distinct valid hypotheses as sampling progresses. In contrast, non-reasoning models plateau earlier, reflecting convergence to smaller hypothesis subsets. The information gain analysis (Figure 4b) further illuminates these exploration dynamics: reasoning models maintain higher marginal information contributions per query, while non-reasoning models exhibit flatter curves characteristic of diminishing returns. This pattern directly correlates with the Recovery Rate degradation observed in Tables 1-3, providing mechanistic evidence that mode collapse manifests as premature entropy saturation rather than uniform exploration inefficiency.

## G. Failure mode decomposition and coverage analysis

Figure 5 decomposes the sources of limited hypothesis coverage through complementary analyses of failure types and exploration completeness. The error categorization (Figure 5a) reveals that mode collapse stems primarily from du-

plicate generation rather than validity failures: while parse errors and constraint violations remain relatively low across models, exact and canonical duplicates constitute the dominant failure mode, particularly for non-reasoning models. This pattern indicates that models can generate structurally valid hypotheses but struggle to diversify beyond preferred templates. The exploration analysis (Figure 5b) quantifies this limitation directly, showing that even high-performing models like GPT-5 and Gemini-2.5-Pro explore only 60-70% of enumerated admissible sets, with weaker models covering substantially smaller fractions. The systematic underexploration persists despite models maintaining high Validity rates, confirming that current LLMs exhibit fundamental constraints in mapping complete hypothesis spaces under underdetermination.

