# OpenReview forum: "HypoSpace: A Diagnostic Benchmark for Set-Valued Hypothesis Generation under Underdetermination and Sublinear Coverage Bounds"
_ICML.cc/2026/Conference — ICML 2026 spotlight_

### Official Review · Reviewer_HG2F · 2026-02-18

**Soundness:** 3
**Presentation:** 2
**Significance:** 3
**Originality:** 3
**Overall Recommendation:** 4
**Confidence:** 3

**Summary:**

HypoSpace is a benchmark across three scenarios of increasing difficulty (1-3) that assesses how LLMs used for hypothesis generation (in scientific discovery) actually sieve the entire hypothesis space rather than fixate on a subgroup, a phenomenon the authors call "mode collapsing". The authors show that most LLMs prefer to generate valid hypotheses rather than achieve greater coverage (as shown by NR and RR metrics). In the end, you can see that reasoning models tend to perform better than non-reasoning ones, and stacking the decoding process improves performance across the board. One thing that I don't understand is what now? Yeah, HypoSpace discovers this phenomenon, but what are the next steps? I feel like the conclusion reiterates the material from the introduction, and the future directions don't address this feeling.

**Compliance With Llm Reviewing Policy:**

Affirmed.

**Final Justification:**

I discussed the authors' work twice during the rebuttal period. They addressed my concerns specifically for the mode collapse issue. I bumped a score to weak accept.

**Key Questions For Authors:**

1) What is this N that you draw for each difficulty? Is it the same across models within the same difficulty? Please provide this information.

2) At the beginning of Section 7.4, you state that because RR is decreasing, there is direct evidence for mode collapse in hypothesis generation. This might be an overstatement. From the tables alone, one can't appreciate this "mode collapse". Can you offer any qualitative examples that might support your claim?

3) I'm curious to see boxplots instead of the Tables 1-3, because I have the feeling that these averages are very shifted due to outliers here. For instance, the higher end of claude-opus-4 in Table 1 for VR at difficulty 1 is 89.30 + 21.70, which suggests there should be an outlier here. Please report boxplots instead of tables, for a complete and fairer picture. It'd be wise to include them in the camera-ready. My suggestion is to report only the hardest setting in the main paper, and the full pictures (i.e., all difficulties) in the appendix.


### Suggestion: It'd be cool if there were a takeaway message at the beginning of the paper (maybe a textbox) that says what the reader should expect from this paper. For instance, *"Exploring the entire space of hypotheses is beneficial for [...] instead of a single hypothesis only."* This could bring more value to the paper. You can take it as you wish. This won't affect the paper's scoring.

**Limitations:**

No, the authors didn't discuss the limitations of their study.

### The following might not be a limitation suggestion but a generic motivation one. It'd be great if the paper included a Motivation section.

I'd expect something along the lines of providing an argument for why exploring all hypotheses is worthwhile. Why does this matter? Is the non-exploration of this hypothesis space really that bad? Based on my experience in counterfactual explainability, if we treat a counterfactual as a hypothesis, we prefer the one closest to the input. Isn't this similar? If you have an observation and a hypothesis, you'd want the hypothesis to be valid and as close as possible to the observation. I mean, some methods produce batches of counterfactuals, but I can't seem to find a reason to explore the entire hypothesis space.

**Strengths And Weaknesses:**

# Soundness
This paper, at first glance, appears interesting, with its claims supported by experiments, specifically Tables 1-3. However, upon closer inspection of these tables, I noticed that it's a bit weird to see that GPT-5, for example, has a standard deviation of 0. I noticed that the captions of these tables emphasize that the reported metrics are averaged across the instances (randomly selected N of them from their generated datasets).

According to the authors, they expect that LLMs will sacrifice NR and RR for VR; however, from the aforementioned tables, for example, non-reasoning models and sometimes reasoning models as well (e.g., claude-opus-4 in Table 1 with difficulty 3, and Table 2) have VR < NR. Obviously, this doesn't invalidate the claims the authors make and try to show theoretically; however, there's no discussion on these cases. What I'm trying to say is that the paper is great at the beginning, but its execution is a bit lacking. I'll also check my reviewer colleagues and see whether they think the same.

# Presentation
In Figure 1(b), my first question was "How do you measure the complexity (settings) of the hypothesis space?" It'd be better if you addressed this now, not just when you talk about the HyroSpace Construction.

I have the impression that the experimental section feels rushed. I would've expected a bit more discussion around the findings (especially Tables 1-3). This could be included if Section 6 is moved to the Appendix (I really don't understand why it's there: does it add anything to the paper?).

Additionally, it's a bit weird that you report the best strategy in red, and the second-best in orange: I would've used bold and underline in the tables. Anyway, if you do boxplots (see my comments in the QUESTIONS section), the best strategy should become clearer. I argue that reporting boxplots is fairer for discerning between performances, given the effects of outliers, compared to reporting "real" ones.

There are plenty of notational mismatches:

* Why are $F_{LLM}$ and $F_\tilde{G}$ different? The notation here is so heterogeneous that I can't follow. Lines 251--258
* Section 5.2: How come here, there's no formalization of validity as in the example for Causal Inference from Perturbations?
* Section 5.3: Again, here, what is $f_\tilde{h}$?
* Where does this $\alpha$ come from in Eq. 6?
* It would be wise that in line 218, you use the same notation as in Eq.5.

Personally, I don't like short related work and deferring the rest to the Appendix, but it's just a matter of taste.

# Significance
I believe the paper addresses a very interesting problem that could pave the way for planning algorithms, as a multitude of hypotheses can be explored for a single observation. Nevertheless, I fail to see why this is better than simply having a single valid hypothesis that is the "best one" according to a specific metric.

# Originality
Yeah, this paper is original since it surfaces a phenomenon which might not have been known in the literature. I tried running the code provided, but couldn't run it for LLaMA since it's the only open-source model (I didn't want to put OPENAI API keys there). Could the authors guide me through this during the rebuttal, please? I'm curious to see what these hypotheses look like qualitatively.


### To the authors: I'm just rating this paper with a reserved rating. If I get my concerns addressed, I'm changing my scores towards acceptance. So please let's have a ping-pong-like discussion during the rebuttal. I really like the paper in the beginning. I just think its execution isn't there yet.

---

> ### Author Rebuttal · Authors · 2026-03-31
>
> **Soundness 1.** The 0.0 standard deviation is not an evaluation artifact; it reflects a ceiling effect. For each difficulty level, we evaluate 30 randomly sampled ground-truth instances and report the mean/std across those instances. In some simpler settings, GPT-5 achieved exactly the same score on all 30 instances (e.g., 100% VR/NR/RR), so the standard deviation is exactly 0.0. This disappears in harder settings (e.g., Boolean), where GPT-5 already shows non-zero variance. We will clarify this in the captions / experimental description.
>
> ---
>
> **Soundness 2.** We agree that the $(NR > VR)$ cases deserve explicit discussion. This follows directly from the metric definitions: VR measures the fraction of valid proposals, whereas NR measures the fraction of unique proposals regardless of validity. Thus, $(NR > VR)$ means that a model is generating outputs that are diverse, but many are invalid under the task constraints. This is exactly why all three metrics are needed: high VR with low NR/RR indicates conservative repetition, whereas $(NR > VR)$ indicates diversity without sufficient validity; RR is necessary because it measures valid and unique discoveries jointly.
>
> ---
>
> **Presentation / Question 3.** We will revise it accordingly: define task complexity earlier (Introduction / Figure 1 caption), shorten Section 6 to free space for more discussion of Tables 1-3, replace red/orange color with standard bold/underline formatting, add boxplots (Anonymous link: https://anonymous.4open.science/r/Fig_boxplots-FB76.), and perform a full notation audit $F_{\mathrm{LLM}}$, task-specific forward functions, missing definitions in Sections 5.2/5.3, $f_h$, $\alpha$, and equation-level consistency). We will also move more related work back into the main text.
>
> ---
>
> **Significance / Limitation.** In settings such as counterfactual explanation, there is often a natural preference ordering (e.g., minimal perturbation), so returning a single “best” hypothesis is reasonable. Under underdetermination, however, multiple mechanistically distinct hypotheses may be equally consistent with the same observations, with no principled way to rank one as uniquely best. This is why broader exploration matters: if a model returns only one valid hypothesis, the user may never realize that equally valid alternatives exist. In scientific reasoning, this matters because follow-up experiments are often designed specifically to distinguish between competing explanations. For this reason, HypoSpace measures Recovery in addition to Validity. Our point is not that every application requires exhaustive enumeration, but that in underdetermined scientific reasoning, evaluating only a single valid answer misses a central part of the problem.
>
> ---
>
> **Originality.** One source of confusion is that, in our current implementation, all models (including LLaMA-3.3-70B) are accessed through API backends rather than local inference, so reproducing the exact setup currently requires API access. We will make this clearer in the documentation and add a cleaner interface for swapping in a local inference backend in the public release. Regarding qualitative behavior, our aggregate failure-mode analysis (Fig. 5) shows that weaker models such as LLaMA-3.3-70B exhibit limited exploration together with substantial duplicate generation and invalid outputs, which is exactly what their low VR/NR/RR reflects. We will add one concrete qualitative example in the appendix.
>
> ---
>
> **Question 1.** $N$ is the number of samples/queries drawn for a given instance. In our main experiments, we set $N=|H_O|$, i.e., the sampling budget equals the size of the admissible hypothesis set for that specific observation set. Thus, $N$ is instance-dependent but model-independent: for the same instance, all models receive exactly the same $N$, while within a difficulty level $N$ can vary across instances because $|H_O|$ varies.
>
> ---
>
> **Question 2.** We agree that declining RR alone should not be described as direct evidence of mode collapse, and we will revise this wording. What RR shows is reduced coverage; stronger evidence comes from inspecting the generated outputs themselves. In the larger-scale Causal setting (see our response to Reviewer U8pD, Tables R1-R2), frontier reasoning models retain 100% Validity, and the RR decline is driven primarily by repeated generation of the same valid structures rather than by invalid outputs. In Boolean (Hard), the pattern is more heterogeneous: non-reasoning models are dominated by exact duplication, whereas reasoning models more often fail through constraint violations and invalid hypotheses, with canonical duplication substantial only for some models. Together, these analyses suggest that the RR decline reflects identifiable output-level patterns, such as duplication, constraint violations, and invalid hypotheses, rather than a purely undifferentiated drop in average performance.

---

> > ### Author Rebuttal · Reviewer_HG2F · 2026-04-03
> >
> > Cool! I would definitely include the boxplots you provided in the main body of the paper (I believe you have an extra page in the camera-ready).
> >
> > Apparently, none of my fellow reviewers noticed the point I raised in **Soundness 2**, but I believe the authors have tried to explain it, although a bit confusingly. This doesn't matter, though. I fully understand that sometimes theory and execution may have exceptions that don't undermine the paper.
> >
> > I'm a bit puzzled by your answer in **Question 2**. Could you reiterate? Is there mode-collapse or not? I couldn't decipher the tables you mentioned in the U8pD's response.

---

> > > ### Author Response · Authors · 2026-04-03
> > >
> > > Thank you for the follow-up. We apologize for the unclear phrasing in our earlier response due to space limitations. The short answer is: yes, mode collapse is present, but its role differs across settings. In some regimes (e.g., larger-scale Causal/3D, and Boolean Hard for weaker models), it is the dominant failure mode; in others (especially Boolean Hard for frontier reasoning models), it is only one component of a broader failure pattern. To clarify our position directly:
> > >
> > > 1. Mode collapse is the dominant failure mode in the following settings:
> > >    - In the larger-scale Causal and 3D settings, **frontier reasoning models** remain 100% valid, and the RR drop comes mainly from repeatedly generating the same valid structures. This is strong evidence of mode collapse.
> > >    - In Boolean (Hard), **weaker/non-reasoning models** also show mode collapse through exact duplication.
> > >
> > > 2. Mode collapse is NOT the dominant failure mode in the following setting:
> > >    - In Boolean (Hard), **stronger reasoning models** often fail through a mixture of constraint violations, invalid hypotheses, and some canonical duplication.
> > >
> > > These observations show that declining RR should not always be described as direct evidence of mode collapse. In some settings, it reflects mode collapse; in others, it reflects a broader reduction in coverage caused by multiple failure modes. We will revise Section 7.4 to make this distinction explicit and avoid overstating RR decrease itself as direct evidence of mode collapse.
> > >
> > > As the rebuttal format this year does not allow for multiple rounds of discussion, we want to sincerely thank you for pushing us to clarify this distinction. If any ambiguity remains, we will make sure it is stated explicitly in the revised version.

---

### Official Review · Reviewer_othL · 2026-03-08

**Soundness:** 2
**Presentation:** 2
**Significance:** 2
**Originality:** 3
**Overall Recommendation:** 4
**Confidence:** 4

**Summary:**

This paper presents HypoSpace, a diagnostic benchmark designed to evaluate how well LLMs generate hypotheses in scientific settings where the evidence does not point to a single answer, specifically where several explanations may fit the same observations. Rather than judging models by whether they produce one correct response, HypoSpace treats them as samplers over finite hypothesis spaces that can be fully enumerated. Their behavior is measured using three metrics: VR, NR, and RR. The authors also provide a theoretical analysis showing that when a model’s generative distribution becomes highly peaked, coverage collapses even if the hypotheses remain valid. The paper introduces a training-free stratified decoding method which partially reduces the coverage collapse.

**Compliance With Llm Reviewing Policy:**

Affirmed.

**Final Justification:**

The paper introduces a useful benchmark, and the rebuttal makes a genuine effort to address the concerns, improving clarity and aligning some claims more closely with the evidence. The additional experiments help, but key issues around generality, and the role of stratified decoding remain only partially resolved. Overall, my assessment has not fundamentally changed, though I appreciate the authors’ thoughtful responses. I have increased my score while still maintaining some reservations.

**Key Questions For Authors:**

Please see the weaknesses section for the key questions.

**Limitations:**

The authors do not mention any limitations in their paper.

**Strengths And Weaknesses:**

# **Strengths**
- The use of deterministic validators and exactly enumerable hypothesis spaces is a major strength. Because admissible hypothesis sets are exactly enumerated, the benchmark avoids the common LLM-as-judge problem. Coverage failures become measurable quantities rather than subjective judgments.

- The experimental setup is also well designed. 7 models are evaluated, covering both reasoning and non-reasoning categories. Each task includes 3 difficulty levels, results are reported as mean ± sd, and a real-world alignment study in Section 8 tests whether the findings extend beyond synthetic setting.

- The failure mode analysis in Appendix G (Figure 5) is really insightful. It shows that coverage collapse mainly comes from duplicate generation rather than invalid outputs or parsing issues. I would suggest the authors to move this analysis/summary of this finding to the main paper.

- The paper addresses a real gap in LLM evaluation where most benchmarks focus on single-answer correctness, while the ability to explore multiple plausible explanations is largely ignored.

- This is practically important, as in scientific reasoning tasks, producing a single valid explanation may be less useful than mapping out the full set of plausible alternatives and the benchmark captures this distinction clearly.


# **Weaknesses**

- From my understanding, the sampling protocol sets choice $(N = |H_O|)$ works well for analyzing coverage but may not reflect practical usage, and it favors tasks where $(|H_O|)$ is relatively small. The paper does not explore how results would change under different budgets. I would suggest the authors to explore this further.
- The authors mention: "Empirically, frontier models confirm this prediction: they maintain high Validity but exhibit limited Uniqueness and Recovery as $|H_O|$ grows". However from table 1 we can see that for the strong models (gpt-5, gemini-2.5-pro and grok-4), the NR and RR values are quite high. Thus this prediction is not supported by the data. For table 2 we see similar patterns for these strong models. The only exception for this is results shown in Table 3 for the DNA task. Thus mentioning this in the paper is not very justified, since it is not supported by the evaluation results. That statement might be true for some of the models, but not for all and thus it is not a very strong claim. Speficially in some cases the NR value is higher than the VR value (table 2, difficulty 2). This basically downplays majority of the claims made in this paper. This statement might be always true for non-reasoning models, but writing it as a general claim is not justified.
- The stratified decoding in Table 4 produces mixed results. In some cases it helps, but for strong models it can hurt performance substantially, for example DeepSeek-R1, GPT-5, etc. The paper notes the variability but does not offer a clear explanation, which weakens the claim that this method meaningfully addresses the problem. Furthermore, the way in which we are getting a lot of new LLMs recently, it becomes challening to figure out if the decoding method would actually help or not.
- Several important analyses appear only in the appendix. At least the failure-mode analysis and the information gain curves (or their summaries) seem central to the paper’s argument and would benefit from appearing in the main text. I would suggest the authors to move these analyses to the main text.
- Table 4 reports the decoding results only for the Boolean interaction task. It is unclear whether similar trends hold for the other tasks, which limits the scope of the proposed mitigation.
- As per my understanding, the three tasks that the paper uses are all combinatorial or symbolic and it does not introduce new architectural or training advances for LLMs, and the only methodological proposal is the heuristic stratified decoding baseline. Although the decoding approach is interesting, its inconsistent effectiveness across different LLMs limits its practical value. With this being one of the main contributions of the paper, the main contribution seems to be more diagnostic rather than corrective.
- The colors used in the figures make it very difficult to understand the figures. I would suggest the authors to use a better color scheme.

---

> ### Author Rebuttal · Authors · 2026-03-31
>
> **Weakness 1.** We agree that exploring different budgets is important. We therefore ran additional experiments on Boolean (Hard), varying the budget from $(1.0\times |H_O|)$ to $(3.0\times |H_O|)$ across three representative models (see our response to Reviewer ER79, Table R1). Larger budgets do not reliably improve Recovery: RR stays nearly flat for GPT-4o and Claude-Opus-4, and is non-monotonic for Gemini-2.5-Pro. At the same time, Novelty drops sharply as budget increases (e.g., Gemini-2.5-Pro: 48.0% $\rightarrow$ 17.1%), indicating that extra samples are mostly duplicates rather than new valid hypotheses. This is consistent with Section 4: under peaked hypothesis distributions, increasing $(N)$ yields diminishing returns because models repeatedly sample the same high-probability modes. Thus, $(N=|Ho|)$ is not artificially restrictive in our setting.
>
> ---
>
> **Weakness 2.** We agree that our original wording was too broad. A more precise reading of Tables 1-3 is that degradation depends on both model capability and hypothesis-space size: at the current Causal/3D scales, the strongest reasoning models remain near ceiling, whereas the effect is already clear for weaker models and in the Boolean task (Table 3). We will revise the paper accordingly, replacing the blanket claim with this more qualified statement.
>
> To test whether the same pattern emerges for frontier models at larger scales, we ran additional experiments on substantially larger admissible sets (see our response to Reviewer U8pD, Tables R1-R2). At 160 ground truths (Causal) and 125 ground truths (3D), all three frontier models show clear NR/RR degradation while retaining 100% VR. This suggests that the effect extends to frontier models, but only at larger $(|H_O|)$.
>
> Regarding NR > VR (e.g., Table 2, difficulty 2), this follows directly from the definitions: VR measures the fraction of valid proposals, whereas NR measures the fraction of unique proposals regardless of validity. Thus, NR > VR indicates diverse but often invalid hypotheses. This is exactly why RR is necessary: it measures valid and unique discoveries jointly.
>
> ---
>
> **Weakness 3.** We agree that the framing of stratified decoding can be more precise. Our main contribution is the diagnostic benchmark, not stratified decoding as a standalone method; Section 6 is intended only as a proof-of-concept showing that changing the sampling distribution can affect recovery behavior.
>
> Table 4 suggests that stratified decoding is model-dependent, not a general remedy. It can help models with very weak baseline exploration (e.g., GPT-4o, LLaMA-3.3-70B, Claude-Opus-4), but it can hurt stronger models such as GPT-5, Gemini-2.5-Pro, and DeepSeek-R1. At the same time, it provides one consistent signal: baseline recovery on complex hypotheses is 0% for all models, while stratified decoding yields non-zero gains for several of them. We therefore view it as evidence that reshaping the sampling distribution can sometimes unlock parts of the hypothesis space that unconstrained decoding misses, rather than as a universal mitigation.
>
> For new LLMs, one cannot reliably predict a priori whether stratified decoding will help. This is precisely where HypoSpace is useful: one can directly compare baseline and stratified decoding on VR/NR/RR to determine whether the intervention improves exploration, only helps within certain complexity regimes, or instead hurts validity and recovery.
>
> ---
>
> **Weakness 4 / 7.** We will revise the paper accordingly: move concise summaries of the failure-mode and information-gain analyses into the main text, and improve figure readability.
>
> ---
>
> **Weakness 5.** We focused Table 4 on Boolean because it is the most discriminative domain in the current benchmark: complexity bias and coverage collapse are strongest there, making it the clearest setting for testing whether distribution reshaping can change recovery behavior. For Causal and 3D, frontier models are already near ceiling under standard decoding, leaving little room for improvement. Additional stratified-decoding experiments on weaker models outside the Boolean domain produced mixed results: they sometimes improved recovery on simpler structures, but did not consistently improve overall or complex recovery.
>
> ---
>
> **Weakness 6.** We agree that the paper is diagnostic rather than corrective, and this is the intended framing. Our main contribution is a benchmark that exposes a missing evaluation dimension: set-valued hypothesis exploration under underdetermination, rather than a new architecture or universally effective decoding method.
>
> This is also why we use combinatorial/symbolic domains: they allow exact enumeration of $(|H_O|)$ and deterministic validation, which are necessary for objective coverage measurement without LLM-as-judge scoring. Stratified decoding is included only as a simple probe to show that changing the sampling distribution can affect recovery behavior, not as a main methodological contribution.

---

> > ### Author Rebuttal · Reviewer_othL · 2026-04-02
> >
> > The added experiments, especially those on budgets and larger hypothesis spaces, help make the recovery–novelty tradeoff easier to see. Removing the earlier claim about frontier models also feels correct. Framing the benchmark as more diagnostic, and pointing out that stratified decoding depends on the model, lines up better with what the results actually show.
> >
> > Still, some of the main issues haven’t gone away. The paper is still a bit too broad in its claims, and properly reflecting the more scale-dependent behavior would need more than small adjustments. It’s reasonable to highlight Boolean as the most discriminative setting, but when performance in other domains is already near the ceiling, it’s hard to tell how much the benchmark adds outside of the harder cases.
> >
> > Stratified decoding also remains somewhat unclear, especially in terms of when it actually helps. The rebuttal clarifies that stratified decoding is meant as a proof of concept rather than a central contribution. But the way it’s presented in the paper still gives it a lot of weight.
> >
> > While the rebuttal does make the presentation clearer, the main concerns still remain. That said, I’m willing to raise the score in light of the authors’ responses.

---

> > > ### Author Response · Authors · 2026-04-03
> > >
> > > We appreciate the reviewer raising the score and the constructive feedback. We will address the two remaining concerns in the revision:
> > >
> > > 1. **Precision of claims.** We will make the following targeted changes: (i) replace the abstract’s “frontier LLMs exhibit a consistent failure mode” with a scale-conditioned statement, namely that all models exhibit degrading Uniqueness and Recovery as difficulty grows, with the onset depending on model capability; (ii) add a paragraph at the end of Section 7.2 explicitly acknowledging that Causal and 3D at current scales are near ceiling for frontier reasoning models, and that Boolean is the most discriminative task in the current benchmark; and (iii) integrate the scaling results (Tables R1--R2) into the main text to provide direct evidence for the extended claim.
> > >
> > > 2. **Stratified decoding weight.** We will condense Section 6 to a short paragraph in the main text that frames it as a diagnostic probe, and move the detailed analysis to the appendix. The reclaimed space will be used to expand the results discussion in Section 7.
> > >
> > > Thank you again for the careful reading and constructive suggestions. We would be happy to clarify any remaining points.

---

### Official Review · Reviewer_ER79 · 2026-03-11

**Soundness:** 2
**Presentation:** 3
**Significance:** 2
**Originality:** 3
**Overall Recommendation:** 4
**Confidence:** 3

**Summary:**

This paper proposes HypoSpace, a diagnostic tool for LLMs in hypothesis generation in underdetermined settings, i.e., when multiple hypotheses can be consistent with the observations. The authors theoretically analyze the limitations of highly concentrated sampling distribution in covering multiple consistent hypotheses. Then, the authors benchmark the validity, uniqueness and recovery rates of state-of-the-art LLMs, showing the strong performance of strong reasoning models, as well as the issue of recovery collapse with difficult problems.

**Compliance With Llm Reviewing Policy:**

Affirmed.

**Final Justification:**

The authors addressed my concerns

**Key Questions For Authors:**

1. Could you discuss more on how this proposed diagnostic tool can help practical use? For example, how does the diagnosis map to the expected behavior of LLMs when used in scientific discovery?
2. It seems that the performance correlates a lot with model abilities, which is not very surprising. Are there more specific takeaways from the benchmarking efforts?
3. How would the benchmark work for noisy real scientific discovery settings?

I'd be open to adjusting the scores if the authors address these questions convincingly.

**Limitations:**

yes

**Strengths And Weaknesses:**

Strengths:

1. The studied problem is interesting and potentially relevant as a controlled sandbox for real scientific discovery.
2. The benchmarking method is novel and might inspire more developments in this direction.
3. The chosen domains are relevant and important.

Weakness:

1. The evaluated domains, while relevant and controlled, seem a bit "toy" as they mostly cover problems like logic deduction, while real scientific discovery deals with noisy data, and exhaustive hypotheses generation might lead to the issue of overfitting to data.
2. The discussion on the usage of this diagnostic toolkit can be strengthened. For example, how does the diagnosis map to the expected behavior of LLMs when used in scientific discovery.
3. The insights from the benchmark could be strengthened. It seems that the performance correlates a lot with model abilities, which is not very surprising. More discussion would be helpful.

---

> ### Author Rebuttal · Authors · 2026-03-31
>
> **Weakness 1 / Question 3.** We agree that real scientific discovery involves noisy data and richer hypothesis spaces. HypoSpace is not intended to simulate end-to-end discovery, but to isolate one specific capability: set-valued inference under underdetermination. This abstraction is deliberate: exact enumeration and deterministic validation make coverage measurable without relying on subjective LLM-as-judge scoring.
>
> In noisy scientific settings, the benchmark would require replacing the binary validity check $(val_O(h)\in\{0,1\})$ with a graded likelihood or score $(P(O\mid h))$, together with a threshold for admissibility. This would make evaluation less objective and blur the boundary between valid and invalid hypotheses. We therefore view noise-robustness and hypothesis-space exploration as complementary but separable challenges, and HypoSpace is designed to isolate the latter. In this sense, HypoSpace also serves as a controlled lower-bound diagnostic: if models already struggle to explore admissible hypothesis sets in fully specified, noiseless settings, this challenge is unlikely to disappear once additional real-world noise and ambiguity are introduced.
>
> We agree that in noisy settings, enumerating all data-consistent hypotheses may also surface noise-fitting explanations. This is why HypoSpace isolates noiseless underdetermination and focuses only on exploration of the admissible set.
>
> Finally, our yeast case study (Sec. 8) shows that underdetermination is not limited to synthetic tasks.
>
> ---
>
> **Weakness 2 / Question 1.** HypoSpace is intended as a diagnostic layer for LLM-assisted discovery pipelines, and its metrics map naturally to practical decisions in three ways.
>
> First, it supports **model selection**: a model with high VR but low RR may be suitable when users only need a few plausible hypotheses, whereas a model with stronger RR is preferable when broader exploration matters. This distinction is useful because standard correctness-style evaluation would treat both models similarly if they produce valid outputs, while HypoSpace reveals whether they systematically miss admissible alternatives.
>
> Second, it supports **diagnosis-to-action mapping**: different metric patterns point to different bottlenecks. Low VR suggests a need for stronger constraint prompting or external validation; high VR with low NR suggests repetitive generation and motivates reject-duplicate sampling or stratified decoding; and persistent RR deficits despite reasonable VR suggest insufficient exploration, motivating ensembling or explicit diversity prompting.
>
> Third, it supports **risk assessment before deployment**: low RR indicates a substantial risk of missing admissible alternatives, which is particularly important in scientific settings where overlooked mechanisms or causal explanations may matter. In this sense, HypoSpace does not replace downstream evaluation, but helps quantify when additional sampling, validation, or human oversight is needed.
>
> ---
>
> **Weakness 3 / Question 2.** We agree that stronger models generally perform better, but the benchmark reveals several more specific takeaways that are not captured by overall capability alone.
>
> First, **validity and coverage are strongly decoupled**: models can remain perfectly valid while still covering only a limited fraction of the admissible set. In our larger-scale follow-up settings (see our response to Reviewer U8pD, Tables R1--R2), GPT-5, Gemini-2.5-Pro, and Grok-4 all retain 100% VR, yet their RR still drops substantially as $(|H_O|)$ grows. This means that standard correctness-style evaluation can overestimate a model’s usefulness for scientific exploration.
>
> Second, **models fail in qualitatively different ways**. Our failure decompositions suggest that limited Recovery does not arise from a single source: depending on the model and task, it can result from duplicate generation, invalid hypotheses, or constraint violations. This suggests that different models may require different interventions, rather than a single universal fix.
>
> Third, **neither simple decoding changes nor larger budgets reliably solve the problem**. Our additional temperature/top-p sweeps (see our response to Reviewer U8pD, Table R3) and budget ablations (Table R1) show that increasing randomness or sampling more hypotheses does not consistently improve RR; instead, models often generate more duplicates or lose validity. This suggests that the core issue is not merely insufficient sampling, but a structural bias toward narrow regions of the admissible hypothesis space.
>
> | Model | Budget | VR (%) | NR (%) | RR (%) |
> |---|---:|---:|---:|---:|
> | GPT-4o | 3.0×\|Ho\| | 92.0 ± 9.6 (+6.4) | 6.1 ± 2.0 (-9.4) | 12.6 ± 4.5 (-1.4) |
> | Claude-Opus-4 | 3.0×\|Ho\| | 36.9 ± 25.0 (-2.6) | 8.5 ± 4.1 (-16.1) | 25.6 ± 12.4 (+1.8) |
> | Gemini-2.5-Pro | 3.0×\|Ho\| | 34.0 ± 0.6 (-20.3) | 17.1 ± 2.9 (-30.9) | 37.3 ± 2.7 (-9.9) |
>
> *Table R1. Performance under different hypothesis budget scales (N).*

---

> > ### Author Rebuttal · Reviewer_ER79 · 2026-04-03
> >
> > Thank you for the response! I raise my score to 4.

---

> > > ### Author Response · Authors · 2026-04-07
> > >
> > > We sincerely thank you for the update and for your careful review. We truly appreciate your feedback and are glad our rebuttal helped clarify the main concerns.

---

### Official Review · Reviewer_U8pD · 2026-03-13

**Soundness:** 3
**Presentation:** 3
**Significance:** 3
**Originality:** 3
**Overall Recommendation:** 5
**Confidence:** 3

**Summary:**

In this paper, the authors introduce HypoSpace, a diagnostic benchmark that examines the quality of large language models (LLMs) as samplers over a finite space of hypotheses in underdetermined situations. The authors employ a set of metrics: Validity (appropriateness), Uniqueness (non-repetition), and Recovery (the fraction of the space that is covered), and apply these metrics to three exactly enumerable domains: causal graph inference from interventions, gravity-limited 3D voxel reconstruction from top-down views, and Boolean genetic interaction modeling. The authors propose a simple theoretical rationale for the phenomenon of reduced recovery with a highly peaked hypothesis distribution. The authors also show that a training-free complexity-aware decoding approach partially mitigates the recovery loss.

**Compliance With Llm Reviewing Policy:**

Affirmed.

**Final Justification:**

The additional experiments on larger hypothesis spaces (Tables R1-R2) and the temperature/top-p sensitivity analysis (Table R3) directly address my concerns. The clarification on the yeast case study is helpful.

**Key Questions For Authors:**

1. How sensitive are RR/NR results with respect to temperature and top-p parameters?
2. For the yeast problem, for the case when $|H_O| > 100$, what were the values of $N$, the total cost, and the cost per hypothesis? Also, how do the models achieve 100% recovery: is it through sampling with $N=|H_O|$, or is it through near-uniform sampling?

**Limitations:**

yes

**Strengths And Weaknesses:**

Strengths:
1. The authors present a clean, interpretable theoretical explanation connecting peaked hypothesis distributions to sublinear coverage growth.
2. The work introduces a simple, reusable complexity-stratified decoding baseline that operationalizes exploration across structural complexity strata.
3. The evaluation of this work includes a diverse set of state-of-the-art reasoning and non-reasoning LLMs across multiple difficulty regimes with reporting on clear mean and std over instances. It also includes an aligned real-world case study like yeast genetic interactions to demonstrate underdetermination and metric behavior beyond synthetic domains.
4. The story, visuals, and tables are overall very intuitive and accessible, clearly showing the framework structure and trends of the performances.

Limitations:
1. The evaluation appears limited in scale (like for 6-node causal DAGs, 3x3x3 voxel worlds, shallow Boolean depths). Thus, the results may not generalize well to larger and more complex hypothesis languages and scientific fields.
2. The paper relies on the most advanced heavy LLMs with at least 70b parameters, which can affect the reproducibility. It would be better to have a comprehensive documentation for decoding hyperparameters and random seeds. There are no results shown for temperature and top-p hyperparameter sensitivity.

---

> ### Author Rebuttal · Authors · 2026-03-31
>
> **Limitations 1.** We agree that the current settings are controlled and smaller than many real scientific hypothesis spaces. This is deliberate: HypoSpace prioritizes exact enumeration and deterministic validation, enabling exact coverage measurement.
>
> To address scalability directly, we ran additional experiments on larger admissible sets (Tables R1-R2). The same pattern persists: as the hypothesis space grows, Recovery (RR) and Uniqueness (NR) drop substantially even when Validity remains high. In both larger causal and 3D settings, all three frontier models retain 100% VR but show clear RR degradation, indicating that the observed coverage collapse is not an artifact of the original benchmark sizes.
>
> More broadly, HypoSpace is not intended to replicate end-to-end scientific discovery, but to isolate one core dimension: set-valued inference under underdetermination. Our yeast case study (Section 8) further shows that such underdetermination is not limited to synthetic tasks: over 100 valid hypotheses remain consistent with the observations.
>
> | Model | VR | NR | RR |
> |-------|--------------|-----------|---------------|
> | GPT-5 | 100.0% (0% ↓)| 71.2% (28.0% ↓) | 71.2% (28.0% ↓) |
> | Gemini 2.5 Pro | 100.0% (0% ↓)| 72.4% (27.0% ↓) | 72.4% (26.8% ↓) |
> | Grok-4 | 100.0% (0% ↓)| 57.7% (42.1% ↓) | 57.7% (42.1% ↓)|
>
> *Table R1. Larger-scale causal inference (160 ground truths); brackets show drops relative to the hardest causal setting in Table 1.*
>
> | Model | VR | NR | RR |
> |-------|--------------|-----------|---------------|
> | GPT-5 | 100.0% (0% ↓) | 75.2% (23.6% ↓) | 75.2% (23.6% ↓) |
> | Gemini 2.5 Pro | 100.0% (0% ↓)| 64.8% (30.4% ↓) | 64.8% (30.3% ↓) |
> | Grok-4 |100.0% (0% ↓)|  79.2%(20.7% ↓) | 79.2%(20.6% ↓)|
>
> *Table R2. Larger-scale 3D voxel reconstruction (125 ground truths); brackets show drops relative to the hardest 3D setting in Table 2.*
>
> ---
>
> **Limitations 2 / Question 1.** We agree that reliance on proprietary models reduces reproducibility. To mitigate this, we include an open-weight baseline (LLaMA-3.3-70B-Instruct), report exact model snapshots (Table 7), and release all prompts, dataset-generation code, and evaluation code in the anonymous repository. Only the LLM inference step requires API access; all downstream evaluation is deterministic. We will also add a table with all decoding settings and seeds.
>
> All main experiments use temperature = 0.7, top-p = 1.0, max output tokens = 40,960, and instance sampling seed = 33550336. We additionally varied temperature and top-p on Boolean (Hard) across three representative models (Table R3). The same qualitative pattern persists: RR/NR vary only modestly across standard settings, and diversity-oriented decoding does not yield a consistent recovery improvement. This suggests that the observed coverage collapse is not simply an artifact of the chosen decoding hyperparameters.
>
> | Model | Metric | Temp=0.2 | Temp=0.7 | Temp=1.0 | Top-p=0.5 | Top-p=0.8 | Top-p=1.0 |
> |---|---|---:|---:|---:|---:|---:|---:|
> | **GPT-4o** | VR (%) | 76.7 ± 38.9 | 88.2 ± 10.4 | 59.2 ± 43.8 | 87.6 ± 19.0 | 82.5 ± 21.8 | 88.2 ± 10.4 |
> |  | NR (%) | 13.6 ± 7.4 | 14.0 ± 5.8 | 19.4 ± 15.9 | 22.3 ± 14.7 | 12.6 ± 4.5 | 14.0 ± 5.8 |
> |  | RR (%) | 8.6 ± 6.4 | 14.0 ± 5.8 | 10.5 ± 6.5 | 16.5 ± 7.1 | 12.6 ± 4.5 | 14.0 ± 5.8 |
> | **Claude-Opus-4** | VR (%) | 33.3 ± 8.4 | 20.1 ± 10.3 | 41.2 ± 13.7 | 32.9 ± 14.0 | 42.4 ± 11.7 | 20.1 ± 10.3 |
> |  | NR (%) | 23.0 ± 9.8 | 17.8 ± 6.6 | 25.7 ± 10.7 | 20.6 ± 9.8 | 21.3 ± 6.8 | 17.8 ± 6.6 |
> |  | RR (%) | 23.0 ± 9.8 | 17.1 ± 5.7 | 24.9 ± 10.7 | 20.6 ± 9.8 | 21.3 ± 6.8 | 17.1 ± 5.7 |
> | **Gemini-2.5-Pro** | VR (%) | 49.8 ± 8.0 | 52.2 ± 12.0 | 53.2 ± 12.9 | 59.1 ± 17.6 | 49.9 ± 20.1 | 52.2 ± 12.0 |
> |  | NR (%) | 46.8 ± 8.8 | 49.9 ± 13.9 | 51.3 ± 14.3 | 51.2 ± 21.1 | 47.9 ± 20.7 | 49.9 ± 13.9 |
> |  | RR (%) | 46.0 ± 9.6 | 49.9 ± 13.9 | 50.3 ± 15.3 | 47.2 ± 16.7 | 47.0 ± 21.2 | 49.9 ± 13.9 |
>
>
> *Table R3. Sensitivity to temperature and top-p on Boolean Genetic Interaction (Hard).*
>
> ---
>
> **Question 2.** The “over 100 valid hypotheses” refers to the setting with only 6 single-KO observations, where $(|Ho|=108)$. Table 5 instead evaluates the final setting with all 13 observations, where $(|Ho|=7)$, using $(N=7)$ samples/model; total inference cost is under `$1`/model. 100% Recovery is therefore not automatic: the model must still generate all 7 distinct valid hypotheses without duplicates in 7 independent samples. GPT-5 and Grok-4 do so, while Gemini-2.5-Pro and DeepSeek-R1 already show performance drop (RR = 85.7%) despite 100% VR.

---

> > ### Author Rebuttal · Reviewer_U8pD · 2026-04-04
> >
> > The additional experiments on larger hypothesis spaces (Tables R1-R2) and the temperature/top-p sensitivity analysis (Table R3) directly address my concerns. The clarification on the yeast case study is helpful. I raise my score to 5.

---

> > > ### Author Response · Authors · 2026-04-07
> > >
> > > Thank you very much for the careful reading and for the encouraging update. We appreciate your thoughtful feedback and are glad the additional experiments and clarifications addressed your concerns.

---

### Decision · Program_Chairs · 2026-04-30

**Decision:**

Accept (spotlight)

**Comment:**

This is an interesting paper, evaluating the LLMs ability, or lack thereof, in generating multi-valued hypotheses for problems that are naturally underdefined, such as causal discovery, or recovering a 3D system from a 2D projection.

The rebuttal seems to have addressed all reviewer concerns. The authors provided experiments on larger hypotheses spaces as a direct reply to Reviewer U8pD. I disagree with Reviewer ER79's comment that domains are "toy" because this is precisely the point of this paper: Explore the limitations of complex, heavily engineered LLMs on domains we understand and have access to the ground truth. If LLM fails, which it does, in these toy tasks, how much should we trust it in more realistic, safety-critical application domains?

An important comment was brought up by Reviewer HG2F on the interpretation of the metrics. The authors' reply on how to jointly interpret them for a meaningful conclusion should be added to the paper.